



# Million-year-scale alternation of warm-humid and semi-arid periods as a mid-latitude climate mode in the Early Jurassic (Late Sinemurian, Laurasian Seaway)

Thomas Munier[1,2], Jean-François Deconinck[1], Pierre Pellenard[1], Stephen P. Hesselbo[3], James B. Riding[4], Clemens V. Ullmann[3], Cédric Bougeault[1], Mathilde Mercuzot[5], Anne-Lise Santoni[1], Émilia Huret[6], Philippe Landrein[6]

[1] Biogéosciences, UMR 6282, uB/CNRS, Université Bourgogne Franche-Comté, 6 Boulevard Gabriel, 21000 Dijon, France.
[2] ISTeP, UMR 7193, SU/CNRS, Sorbonne Université, 4 Place Jussieu, 75005 Paris, France.
[3] Camborne School of Mines and the Environment and Sustainability Institute, University of Exeter, Penryn Campus, Penryn, Cornwall TR10 9FE, UK.
[4] British Geological Survey, Keyworth, Nottingham NG12 5GG, UK.
[5] Géosciences Rennes, UMR 6118, UR/CNRS, Université Rennes 1, Campus de Beaulieu, CS 74205 35042 Rennes cedex,
France.
[6] Agence Nationale pour la gestion des déchets radioactifs, Centre de Meuse/Haute-Marne, RD 960, 55290 Bure, France.

*Correspondence to:* Thomas Munier (thomas.munier@sorbonne-universite.fr)

**Abstract.** High resolution clay mineral and stable isotope (C, O) data are reported from the upper Sinemurian (Lower Jurassic)
of the Cardigan Bay Basin (Llanbedr [Mochras Farm] borehole, northwest Wales) and the Paris Basin (Montcornet borehole, northern France) to highlight the prevailing environmental and climatic conditions. In both basins, located at similar palaeolatitudes of 30–35°N, the clay mineral assemblages comprise chlorite, illite, illite-smectite mixed-layers (R1 I-S), smectite and kaolinite in various proportions. Because the influence of burial diagenesis and authigenesis is negligible in both boreholes, the clay minerals are interpreted to be derived from the erosion of the Caledonian and Variscan massifs, including
their basement and pedogenic cover. In the Cardigan Bay Basin, the variations in the proportions of smectite and kaolinite are inversely related to each other over the entire upper Sinemurian succession. As in the Pliensbachian, the stratigraphical distribution reveals an alternation of kaolinite-rich intervals reflecting strong hydrolysing conditions, and smectite-rich intervals indicating a semi-arid climate. Kaolinite is particularly abundant in the upper part of the *obtusum* Zone and in the *oxynotum* Zone, suggesting more intense hydrolysing conditions likely coeval with warm conditions responsible for an
acceleration of the hydrological cycle. This interval is also marked by a negative excursion of $\delta^{13}C_{carb}$ and $\delta^{18}O_{carb}$, which may confirm a warmer palaeoclimate, although these excursions may be exaggerated or overprinted by the carbonate diagenesis. In the north of the Paris Basin, the stratigraphical succession is much thinner compared to the Cardigan Bay Basin site, and the *oxynotum* Zone is either absent or highly condensed. The clay assemblages are dominantly composed of illite and kaolinite



without significant stratigraphical trends, but a smectite-rich interval identified in the *obtusum* Zone is interpreted as a
consequence of the emersion of the London-Brabant Massif following a lowering of sea-level. A long-term decrease of $\delta^{13}C_{org}$
from the late *oxynotum*/early *raricostatum* zones is also recorded in the two sites and may precede or partly include the negative
carbon-isotope excursion of the Sinemurian/Pliensbachian Boundary Event, recognised in most basins worldwide, and
interpreted as a late pulse of the Central Atlantic Magmatic Province volcanism.

## 1 Introduction

The Early Jurassic is characterised by major palaeogeographical changes induced by the breakup of Pangaea. This geodynamic
evolution is accompanied by intense volcanic activity corresponding to the Central Atlantic Magmatic Province (CAMP),
beginning at the Triassic/Jurassic boundary ~201.5 million years ago (Marzoli et al., 1999; McHone, 2000; Davies et al. 2017),
and likely responsible for the end-Triassic mass extinction (see e.g. Korte et al., 2019 and references therein). The breakup of
Pangaea led to the opening of the Hispanic and Viking corridors, connecting the Tethys Ocean respectively to Panthalassa and
the Arctic Ocean (Bjerrum et al. 2001; van de Schootbrugge et al., 2005; Damborenea et al. 2013; Porter et al., 2013). The
disintegration of Pangaea resulted in the formation of many sedimentary basins, and palaeogeographical changes led to
exchanges of water masses that triggered climate fluctuations with colder intervals (Dera et al., 2011) over a prolonged
greenhouse period (Chandler et al., 1992; Dera et al., 2009a, 2015; Korte et al., 2015).

Reconstruction of seawater temperatures through the Early Jurassic are mostly deduced from $\delta^{18}O$ measurements of belemnite
rostra and some other mollusc shells, notably oysters. During the Late Sinemurian, oxygen isotope data ($\delta^{18}O_{carb}$ and $\delta^{18}O_{belm}$)
show increasing values over time (Dera et al., 2011) indicating cooler ocean temperatures, recorded for example in the
Cleveland Basin (Hesselbo et al., 2000; Korte and Hesselbo, 2011) or the Asturian Basin (Gómez et al., 2016). However,
warmer conditions seem to have prevailed episodically for example during the *oxynotum* Zone (Riding et al., 2013; Hesselbo
et al. 2020).

The carbon cycle also shows perturbations with negative excursions, recorded either by $\delta^{13}C_{carb}$ or $\delta^{13}C_{org}$. The best documented
of these excursions is the Sinemurian-Pliensbachian Boundary Event (SPBE or S-P Event), recognised in many areas, including
among others: the Cleveland Basin (Hesselbo et al., 2000; Jenkyns et al., 2002; Korte and Hesselbo, 2011), the Wessex Basin
in Dorset (Jenkyns and Weedon, 2013; Price et al., 2016), the Cardigan Bay Basin in West Wales (van de Schootbrugge et al.,
2005; Hesselbo et al., 2013; Storm et al., 2020), the Lusitanian Basin (Duarte et al., 2014; Plancq et al., 2016), the Paris Basin
(Peti et al., 2017; Bougeault et al., 2017), and the Central High Atlas Basin of Morocco (Danisch et al., 2019; Mercuzot et al.,
2019). This negative excursion is recorded in carbonates ($\delta^{13}C_{carb}$), belemnite rostra ($\delta^{13}C_{belm}$), and organic matter ($\delta^{13}C_{org}$).
Another $\delta^{13}C_{org}$ negative excursion was also first recognised in the upper part of the *obtusum* Zone and through the *oxynotum*
Zone in Eastern England (Riding et al., 2013; Hesselbo et al. 2020), and later recorded in Dorset (Southern England, Jenkyns
and Weedon, 2013), in the Mochras borehole (Storm et al., 2020), and in the Southern Alps in Italy from shallow-water
carbonate platforms to deep offshore environments (Masetti et al., 2017). This excursion coincides with increasing proportions
of two thermophilic palynomorph taxa, *Classopollis classoides*, a terrestrially-derived pollen grain, and *Liasidium variabile*,

a marine dinoflagellate cyst, suggesting that the *oxynotum* Zone was a warm and/or dry interval. *Liasidium variabile,* a reliable index for the Upper Sinemurian in northwest Europe (Brittain et al., 2010; van de Schootbrugge et al., 2019), may have invaded the Tethys Ocean from Panthalassa after the opening of the Hispanic corridor (van de Schootbrugge et al., 2005). This species

is particularly abundant in the *oxynotum* Zone, and the name Liasidium Event was used to describe the complex of environmental changes at this time (Hesselbo et al., 2020).

Humidity is also a key parameter of climate, but it is poorly documented over this period. Palynological data are focused on *Classopollis* pollen, which are very common in the *obtusum* and *oxynotum* zones, whether in the Cardigan Bay (Wall, 1965; van de Schootbrugge et al., 2005), the Cleveland (Riding et al., 2013) or the Lusitanian basins (Poças Ribiero et al., 2013).

The composition of clay mineral assemblages can be a reliable climate indicator (Chamley, 1989; Ruffell et al., 2002; Raucsik and Varga, 2008; Dera et al., 2009b) provided their dominant detrital origin can be demonstrated. Clay mineral assemblages may reflect the intensity of hydrolysis during pedogenic processes and runoff conditions on landmasses, and thus specify humidity variations from the signal recorded in marine sedimentary series. In the Upper Sinemurian, variations of clay mineral assemblages have been studied on several outcrops and boreholes from the British Isles (Jeans, 2006; Kemp et al. 2005;

Hesselbo et al., 2020) and in the Montcornet borehole, north of the Paris Basin (Debrabant et al., 1992), but at low resolution, only for a short interval, or in successions affected by strong clay mineral diagenesis. Here we attempt, through a high-resolution study of detrital clay mineral assemblages and fluctuations in stable isotopes (C and O) of Upper Sinemurian strata from the Llanbedr (Mochras Farm) and the Montcornet boreholes, to estimate the intensity of chemical weathering, of hydrolysis, and the magnitude of carbon cycle changes.

## 85   2 Geological background

During the Early Jurassic, the Paris and Cardigan Bay basins were located to the north west of the Tethyan domain. This area corresponded to an archipelago of large islands inherited from Caledonian and Variscan massifs (Thierry et al., 2000). These continental masses, such as the London-Brabant Massif, the Massif Central and Armorican Massif, and the Welsh High, were surrounded by an epicontinental sea (Fig. 1). An excellent sedimentary record of the Early Jurassic is preserved at both study

locations due to the extensional context linked to the breakup of Pangaea and related thermal and tectonic subsidence (Woodland et al., 1971; Corcoran and Clayton, 1999; Guillocheau et al., 2000; Beccaletto et al., 2011; Hesselbo et al., 2013).

### 2.1 The Llanbedr (Mochras Farm) borehole, Cardigan Bay Basin, Wales

The Cardigan Bay Basin, located in West Wales, between the Welsh High and the Irish Massif (Fig. 1), corresponds to a half-graben basin (Dobson and Whittington, 1987; Tappin et al., 1994; Holford et al., 2005), bounded at the end of the Palaeozoic

by a major active fault (the Mochras Fault) with an offset of almost 4500 m (Wood and Woodland, 1968; Woodland, 1971, Hesselbo et al., 2013). This basin was located at a latitude between 35°N and 40°N (Fig. 1; Thierry et al., 2000). Lower Jurassic strata reach ~1300 m in thickness (Woodland et al., 1971; Ruhl et al., 2016), and contrast with overlying thinner Middle and



Upper Jurassic successions (Dobson and Whittington, 1987). Cretaceous strata are rare or absent, because of erosion following thermal uplift and inversion processes (Woodland et al., 1971; Tucker and Arter, 1987; Tappin et al., 1994), while the thickness

of Cenozoic sedimentary rocks reaches 600 m.

The Llanbedr (Mochras Farm) borehole, commonly abbreviated to Mochras, was drilled between 1967 and 1969 on the coast in northwest Wales, UK (Fig. 2) (Woodland et al., 1971; Hesselbo et al., 2013). The Lower Jurassic deposits consist of a continuous succession of marls and claystones, making this borehole a reference for environmental and climatic reconstructions for the Early Jurassic (Hesselbo et al., 2013).

The 220 m-thick Upper Sinemurian strata consist of relatively homogeneous marls and clayey mudstones, with more silt in the upper part of the *raricostatum* Zone. Veins of calcite and pyrite commonly occur and siderite nodules are present, particularly in the *oxynotum* Zone. It has previously been suggested that this calcite veined level may have been the consequence of faulting leading to very minor stratigraphical offset according to biostratigraphical data (Woodland et al., 1971; Storm et al., 2020), but there is no positive evidence for strata missing due to tectonism.

Following the zonation proposed by Page (2003), a precise biostratigraphical scheme is established, based on the frequent occurrence of ammonites (Hesselbo et al., 2013; Page in Copestake and Johnson, 2014). The *obtusum* Zone extends from 1468 to 1376 m, overlain by the *oxynotum* Zone to 1332 m, and the *raricostatum* Zone from 1332 to 1249 m (Woodland et al., 1971).

## 2.2 Montcornet borehole, Paris Basin, France

During the Early Jurassic, the Paris Basin was bordered by continental masses, remnants of the Palaeozoic orogenic belts (Fig. 1). The main landmasses include the London-Brabant Massif (LBM), the Armorican Massif, the Massif Central, and the Bohemian Massif (Guillocheau et al., 2000; Thierry et al., 2000).

The Montcornet borehole (borehole Andra A901) was drilled in 1989 near the village of Montcornet to the north of the Paris Basin (Fig. 3), by Andra, the French National Radioactive Waste Management agency. The strata penetrated were deposited

in an epicontinental sea located immediately south of the LBM, at a palaeolatitude of approximately 35°N, quite similar to the latitude of the Cardigan Bay Basin (Fig. 1). The stratigraphical succession extends from the Devonian metamorphic basement (shales) to the Turonian chalk, with a major gap occurring for the Lower Cretaceous, corresponding to the continental evolution of the Paris Basin (Wealden facies), from the Purbeckian deposits to the upper Albian transgressive claystones and marls (Debrabant et al., 1992). Above the siliciclastic continental Triassic deposits, at a depth of ~1075 m, Jurassic strata, nearly 870

m thick, correspond mainly to open-marine limestones and marls (Yang et al., 1996).

The Upper Sinemurian succession (~50 m, Fig. 4) consists of alternations of claystone, marl and bioclastic limestone beds characterised by the presence of bivalves (*Gryphaea)*, likely deposited in lower to upper offshore environments according to Yang et al. (1996).

The biostratigraphical calibration is complicated by the occurrence of some important hiatuses and by the relative scarcity of

ammonites. Some new determinations (by J.L. Dommergues) on ammonites found during the sampling complete the previous





biostratigraphical scheme of Yang et al. (1996) and are illustrated in Fig. 4. Ammonites from the *semicostatum* Zone (Lower Sinemurian) are identified up to 976 m while the first ammonite from the *obtusum* Zone occurs at 948.88 m with *Promicroceras* gr. *planicosta* (Yang et al., 1996, Fig. 4). The *turneri* Zone is not identified, but may be present between 976 m and 948.88 m. However, the occurrence of the dinoflagellate *Liasidium variabile* in this interval strongly suggests a Late Sinemurian age, the

miss of the *turneri* Zone (Fauconnier, 1995; Bucefalo Palliani and Riding, 2000) and that the *obtusum* Zone starts at 976 m. The last ammonite of the *obtusum* Zone is identified at 939.80 m while the first ammonite of the *raricostatum* Zone is identified at 937.82 m (Yang et al., 1996) or slightly lower (938.55 m) according to new determinations of *Crucilobiceras* sp. that would indicate the *densinodulum* Subzone of the *raricostatum* Zone. (Fig. 4). No ammonites from the upper part of the *obtusum* Zone or from the *oxynotum* Zone have been found. This interval is, as is the case in the Wessex Basin in Dorset (Lang, 1945; Hallam

1999; Hesselbo, 2008) and the Lower Saxony Basin of Germany (van de Schootbrugge et al., 2019), either absent or highly condensed in a 1.25 m thick interval without any ammonites between 939.80 and 938.55 m. The last ammonites of the *raricostatum* Zone are identified up to 921.75 m, in agreement with the occurrence of *Liasidium variabile* up to 922 m (Fauconnier, 1995), while the first ammonites of the *jamesoni* Zone appear from 920.6 m, suggesting that the Sinemurian/Pliensbachian boundary is located at ~921 m (Fig. 4). However there are also some gaps at the

Sinemurian/Pliensbachian transition since the upper part of the *raricostatum* Zone (*aplanatum* Subzone) is possibly missing, and since the base of the *jamesoni* Zone (*taylori* Subzone) is also missing.

# 3 Material and methods

## 3.1 Clay mineral analyses

A total of 223 samples were analysed using X-Ray Diffraction (XRD). After moderate grinding in a mortar, powdered samples

were decarbonated with a 0.2N HCl solution. The < 2 µm fraction (clay-sized particles) was extracted with a syringe after decantation of the suspension for 95 minutes following Stokes' Law; this fraction was then centrifuged. Clay residue was then smeared on oriented glass slides and run in a Bruker D4 Endeavor diffractometer with $CuK_\alpha$ radiation, a LynxEye detector and a Ni filter with a voltage of 40 kV and an intensity of 25 mA (Biogéosciences laboratory, University of Burgundy, France). Goniometer scanning ranged from 2.5° to 28° for each analysis. Three runs were performed for each sample to discriminate

the clay phases: 1) air-drying; 2) ethylene-glycol solvation; and 3) heating at 490°C for two hours, as recommended by Moore and Reynolds (1997). Clay minerals were identified using their main diffraction ($d_{001}$) peaks and by comparing the three diffractograms obtained. The following main clay minerals were identified: a R0 type illite-smectite mixed-layer (17 Å based on a glycolated run) referred to as smectite for the following sections; a R1 type illite-smectite mixed-layer (around 11.5 Å in air-drying conditions and 13 Å after ethylene-glycol solvation); chlorite (14.2 Å, 7.1 Å, 4.7 Å and 3.54 Å peaks); illite (10 Å,

5 Å, 3.33 Å peaks) and kaolinite (7.18 Å and 3.58 Å peaks). Each clay mineral was quantified using the MacDiff software (version 4.2.5) (Petschick, 2001) on glycolated sample diffractograms. The area of main peaks ($d_{001}$) is measured to estimate by semi-quantification, the proportion of each clay species. As the main ($d_{001}$) peak of kaolinite and the ($d_{002}$) peak of chlorite



overlap, a deconvolution procedure was applied on the ($d_{004}$) peak area of chlorite (3.54 Å) and ($d_{002}$) peak area of kaolinite (3.57 Å) to accurately quantify both mineral portions using the 7.1 peak ($d_{001kaolinite} + d_{002chlorite}$). Chlorite percentage was

calculated using the mean between ($d_{001}$) and ($d_{002}$) chlorite peak areas considering that the chemical nature of chlorite impacts the ($d_{001}$)/($d_{002}$) ratio (Moore and Reynolds, 1997). The error margin of this method is approximatively ± 5 % on the relative proportions of clay minerals in the clay fraction. The relative proportions of clay minerals are estimated using the ratios between the areas of the peaks, the most relevant of these being the K/I and Sm/K ratios.

### 3.2 Geochemical preparation and analyses

Seventy samples from the Mochras borehole and 38 samples from the Montcornet borehole were selected for $\delta^{13}C_{org}$ analyses. One gram of sample, previously crushed, underwent an acid digestion by 10 ml of 6N HCl solution for 48 hours, a concentration and a duration justified by the common occurrence of dolomite. After cleaning with pure water, decarbonated powders were dried in an oven (50°C for 24 to 48 hours), then crushed again to obtain a fine powder. Each sample, of a specific mass (9 to 75 mg, depending on former sample $CaCO_3$ content), is weighed in a tin capsule with a Sartorius M2P ultrabalance.

Samples were analysed with the Elementar MICRO cube elemental analyser coupled to an Elementar Isoprime 100 isotope ratio mass spectrometer. Isotope ratios obtained were compared to international standards USGS40 L-glutamic acid ($\delta^{13}C$ = -26.39 ± 0.04‰ V-PDB) and IAEA-600 caffeine ($\delta^{13}C$ = -27.77 ± 0.04‰ V-PDB). For each sample, replicates showed reproducibility better than ± 0.15‰. The total organic carbon (TOC) content, expressed in wt.% was determined by the elemental analyser at the same time.

Bulk rock carbon and oxygen isotope ratios analyses coupled with $CaCO_3$ concentration measurements were carried out using a Sercon 20-22 triple detector Gas Source Isotope Ratio Mass Spectrometer at the University of Exeter Penryn Campus following methods described in detail in Ullmann et al. (2020). Bulk rock powder extracted from rock fragments with a handheld drill was weighed at 1 µg precision and transferred into borosilicate vials targeting an amount of 500 µg of $CaCO_3$ for analysis. Samples were flushed with He to remove atmospheric gases and then reacted with nominally anhydrous

phosphoric acid ("103 %") at 70°C. In a single batch 80 samples were analysed together with two in-house standards (22 aliquots of CAR- Carrara Marble, $\delta^{13}C$ = +2.10 ‰ V-PDB; $\delta^{18}O$ = -2.03 ‰ V-PDB; 8 aliquots of NCA- Namibia Carbonatite, $\delta^{13}C$ = -5.63 ‰ V-PDB; $\delta^{18}O$ = -21.90 ‰ V-PDB). Instrumental drift and biases were corrected using a two-point calibration constrained by these two in-house standards. Accuracy was ensured via previous calibration of the in-house standards against international certified standards. $CaCO_3$ content was computed from matching signal intensity of unknowns with CAR, which

is assumed to be 100% pure $CaCO_3$ (44 wt% $CO_2$). Reproducibility of the isotope ratio measurements for Mochras bulk rock samples based on analyses of CAR (2 s.d., n = 300) is 0.07 ‰ for $\delta^{13}C$ and 0.16 ‰ for $\delta^{18}O$. Reproducibility of $CaCO_3$ determinations are based on multiple analysis of NCA as an unknown which gave 97.2 ± 1.3 % (2 s.d., n = 132).

$CaCO_3$ content was measured using a Bernard calcimeter (volumetric calcimetry) on samples, completed by weight-loss method (weight difference between the sample before and after decarbonation was performed prior to isotopic analyses on





organic matter) for each sample. Each curve obtained for the geochemical data has been refined as a smoothed curve and its
95% confidence intervals acquired from a Kernel type regression using different levels of smoothing for each borehole.

## 4 Results

### 4.1 CaCO₃ and Total Organic Carbon (TOC) contents

Calcite content from the Upper Sinemurian of the Mochras borehole shows substantial fluctuations between 3 and 62% (Fig.
5). The Lower Sinemurian/Upper Sinemurian boundary is relatively rich in carbonate, up to 40%, but the lowermost and the
uppermost parts of the *obtusum* Zone are depleted in $CaCO_3$ (<10%), while in the middle part of this ammonite zone the $CaCO_3$
content ranges from 20 to 40%. The low $CaCO_3$ content recorded at the top of the *obtusum* Zone persists in the lower half of
the *oxynotum* Zone. Then, in the upper half of this zone and in the *raricostatum* Zone, $CaCO_3$ increases up to 60% (Fig. 5). In
the Montcornet borehole $CaCO_3$ content also fluctuates between 3 and 62%, with similar trends, notably: 1) a depletion in the
uppermost part of the *obtusum* Zone, although less well expressed than in the Mochras borehole likely because of the
condensation of the series, and; 2) a significant increase in the *raricostatum* Zone (Fig. 6).

Total organic carbon (TOC) measurements are also similar between the two boreholes with proportions around 1% (Figs 7,
8). The Lower Sinemurian/Upper Sinemurian boundary is marked by a slightly higher TOC content (1.5%) while the top of
the *obtusum* Zone shows a decrease in the proportion of organic carbon (0.5%). In the Mochras borehole, the *macdonnelli-*
*aplanatum* subzones of the *raricostatum* Zone are enriched in TOC with values generally higher than 1.5% and reaching 3%.

### 4.2 Clay mineralogy

#### 4.2.1 Mochras borehole

Upper Sinemurian clay mineral assemblages are dominantly composed of chlorite (5 to 32%), illite (15 to 42%), R0 type
illite/smectite mixed-layers hereafter called smectite (10 to 60%) and kaolinite (4 to 32%). Minor proportions of R1 type
illite/smectite mixed-layers are commonly associated with these minerals. Significant fluctuations in the relative proportions
of the different clay species are recorded along the core. The main striking feature is the inverse relationship between smectite
and kaolinite, particularly well-expressed by the Sm/K ratio (Fig. 5). The opposition of these two minerals determines an
alternation of kaolinite-rich packages of sediments (lower part of *obtusum*, top of *obtusum*/base *oxynotum* and median part of
*raricostatum*), and of smectite-rich packages of sediments (middle part of *obtusum*, upper part of the *oxynotum* zone / base
*raricostatum*, and upper part of *raricostatum)*.

The proportions of chlorite are relatively high and fluctuate in parallel with those of kaolinite. From the base to the top of the
Upper Sinemurian, the proportion of illite increases more or less regularly from ~20 to 40%.



### 4.2.2 Montcornet borehole

The clay mineral assemblages of the Upper Sinemurian of the Montcornet borehole are composed of the same minerals as the
Mochras borehole, but show much less variation (Fig. 6). Illite is the most abundant clay mineral with proportions from 25 to
46%, without any clear trend through the core. Kaolinite is also abundant with proportions ranging from 6 to 32%, most
samples having values close to 22%. Chlorite shows lower percentages, between 9 and 24% (average of ~19%). According to
Debrabant et al. (1992), this mineral is associated with small proportions of chlorite/smectite mixed-layers, undifferentiated
on Fig. 6. R1 type illite/smectite mixed layers are relatively abundant between 5 to 26% notably in the middle part of the
*obtusum* Zone. Smectites are absent over a large part of the 56 m of Upper Sinemurian, but these clay minerals occur in
significant proportions (up to 33%) in an eight metre-thick interval between 965 and 973 m within the *obtusum* Zone (Fig. 6).

### 4.3 Carbon and oxygen isotope fluctuations

### 4.3.1 Mochras borehole

Organic carbon isotopes ($\delta^{13}C_{org}$) values show significant variations (about 3.9‰) between -24.47 and -28.34‰ over the Upper
Sinemurian succession (Fig. 7). $\delta^{13}C_{org}$ values show little variations in the *obtusum* and *oxynotum* zones (around -25/-26‰),
while in the *raricostatum* Zone, an irregular decrease of the values down to -28‰ is recorded.
$\delta^{13}C_{carb}$ shows significant variations of more than 5‰ with values between +3.04 and -2.21‰ V-PDB (Fig. 7). A prominent
negative excursion (~ 3‰) is recorded at the transition between the *obtusum* and the *oxynotum* Zones followed by an increasing
trend up to the topmost part of the Sinemurian. A slight negative excursion (1‰) is recorded at the transition between the
*densinodulum/raricostatum* and *macdonnelli/aplanatum* subzones (Fig. 7).
$\delta^{18}O_{carb}$ values range from -6.54 to -2.61‰ (Fig. 7). Very large fluctuations are recorded in the *obtusum* and *oxynotum* zones,
while more constant values around -4‰ are observed in the *raricostatum* Zone. The major part of the *oxynotum* Zone
corresponds to an interval characterised by lower values.

### 4.3.2 Montcornet borehole

The isotope data ($\delta^{13}C_{org}$) from the Montcornet borehole show values between -26.49 and -24.66‰ (Fig.8). The trends are
similar to those of the Mochras borehole, although less well-expressed likely due to the condensation of the series and the
probable occurrence of hiatuses. The values increase slightly from the base of the core to the transition between the *obtusum*
and *raricostatum* zones, while a decreasing trend is observed in the *raricostatum* Zone to the base of the Pliensbachian. The
isotopically lowest values of ~-26.5 ‰ recorded in the *raricostatum* Zone are however higher than at Mochras.



## 5 Discussion

### 5.1 Diagenetic influence

#### 5.1.1 Influence of diagenesis on clay mineral assemblages

The use of clay minerals as climatic proxies assumes that these minerals are mainly of detrital origin. However, the increase
in temperature associated with burial may trigger various transformations of detrital clay minerals to change the constitution
of detrital clay assemblages. In fine-grained clayey and marly sediments, among the possible transformations, the illitisation
of smectite is certainly the most important. The illitisation of smectites into R1-type illite/smectite mixed-layers begins when
the temperatures reach 60-70° C at a depth of burial of the order of 2000 m, considering a normal geothermal gradient (Šucha
et al., 1993; Lanson et al., 2009; Dellisanti et al., 2010).

The occurrence of abundant smectite in the Sinemurian strata in the Mochras borehole and the presence of a smectite-rich
interval in the Montcornet borehole indicate a limited diagenetic influence due to the relatively shallow depth of burial. In both
boreholes, the maximum burial temperatures probably never exceeded 70°C, which is consistent with the geological history
of the Cardigan Bay and Paris basins. In the Cardigan Bay Basin, the thickness of the sediments overlying the Sinemurian can
be estimated to be ~1400 m including 800 m of Lower Jurassic and ca. 600 m of Oligo-Miocene and Quaternary sedimentat

rocks, with, of course, any of the younger Mesozoic strata eroded before deposition of the Cenozoic sediments (Tappin et al.,
1994; Holford, 2005). The negligible influence of burial diagenesis is also confirmed by the occurrence of immature to only
marginally mature organic matter (OM) revealed by Rock Eval pyrolysis data from the Sinemurian mudrocks (van de
Schootbrugge et al., 2005; Storm et al. 2020). In the Sinemurian succession, $T_{max}$ values range between 423 and 436°C (average
428°C, n = 195, Storm et al., 2020) indicated immature OM or occasionally early mature OM at the onset of the oil window.

Vitrinite reflectance ($R_0$ max) data suggest a higher maximal burial temperature of the Sinemurian strata (Corcoran and
Clayton, 1999) ranging between 83 and 90°C (Holford et al., 2005), but these authors discarded the lowest values of $R_0$ (low
burial temperatures), considering that these data were not reliable enough. Such high temperatures are incompatible with both
the occurrence of smectite and with the presence of immature organic matter, and therefore we consider that the low values of
vitrinite reflectance data published by Holford et al. (2005) are fully realistic.

In the Montcornet borehole, the depth of burial of the Upper Sinemurian can be estimated at a maximum of ca. 2000 m,
including the entire Jurassic succession (870 m), ca. 200 m of Cenomanian and Turonian chalks and now eroded/dissolved
Coniacian to Maastrichtian chalks. The Coniacian to Campanian chalk is ~ 400 m-thick in central Paris Basin (southwest of
Paris, Robaszynski et al., 2005) but the uppermost Campanian and Maastrichtian deposits were eroded and therefore the
thickness of the entire Upper Cretaceous is difficult to estimate. To the East of the Paris Basin, a maximum thickness of the

chalk deposits is estimated around only 200 m (Blaise et al., 2014). According to the apatite fission-track thermochronology
study of Barbarand et al. (2018), the entire Cretaceous deposits would have a thickness of 1000 m, which seems to be a
maximum value. Assuming a total burial depth of 2000 m, which represents probably a maximum, the burial-temperature rise
probably did not exceed 60°C, which is consistent with the preservation of smectite. This is also confirmed by $T_{max}$ data which



range between 423 and 426°C in the Sinemurian of the Montcornet borehole, indicating that organic matter is still immature
(Disnar et al., 1996; Mercuzot et al., 2019).

Clay diagenesis can be also revealed by relationships between clay mineralogy and the lithology, notably in marl-limestone alternations (Deconinck and Debrabant, 1985; Deconinck, 1987; Levert and Ferry, 1988), but in the two studied boreholes, there is no statistically strong correlation between $CaCO_3$ content and the proportion of each clay species (Fig. 9). As an example, in the Mochras borehole, the proportion of smectite is weakly correlated with the percentages of $CaCO_3$ (r=0.31, n

= 128, p-value < 0,05) which likely indicates that a possible better preservation of smectite on carbonate-rich interval can be excluded.

The occurrence of authigenic well-crystallised kaolinite can also be envisaged, as it was previously observed in some porous carbonates in the upper Pliensbachian of the Paris Basin (e.g. Bougeault et al., 2017). However, this phase can be easily highlighted on diffractograms by the presence of very narrow peaks indicating a good crystallinity, which is not the case here,

thus excluding the occurrence of measurable authigenic kaolinite. Moreover, sampling of porous limestones has been avoided. Contrary to what has been observed in the overlying Pliensbachian strata of the Mochras borehole, we do not identify any authigenic mineral such as clinoptilolite or berthierine whose presence there in small quantities is likely in the Pliensbachian (Deconinck et al., 2019). Consequently, we infer that most clay minerals identified in the Sinemurian of the two studied boreholes are dominantly detrital and carry climatic and environmental information.


### 5.1.2 Impact of diagenesis on isotopic data

Fluid circulations and temperature may disturb the primary isotope signal in sediment during late diagenesis, notably for the bulk carbonate signal (Anderson, 1969; Hudson, 1977; Marshall, 1992). Early diagenesis may also impact the isotope signal from bulk carbonate when low calcium carbonate content is present (Ader and Javoy, 1998; Bougeault et al., 2017).

$\delta^{18}O$ values are more sensitive to fluid circulation and recrystallisation, particularly in porous rocks, normally leading to more negative and/or scattered values (Hudson, 1977; Marshall, 1992; Stoll and Schrag, 2000). Although the low porosity and permeability of clayey limestones, marls and claystones of the Mochras and Montcornet boreholes are not favourable to fluid circulation, some diagenetic features such as the occurrence of nodular beds, and calcite and siderite nodules, suggest that isotopic values may be significantly altered by diagenetic processes. In the Montcornet borehole, this is the case in the

underlying Hettangian succession where the secondary crystallisation of calcite in sulphate-reducing environments is responsible for a depletion in $^{13}C$ (Ader and Javoy, 1998). In the overlying Pliensbachian strata of this borehole, an alteration of the $\delta^{13}C_{carb}$ was observed by Bougeault et al. (2017), notably in a carbonate-depleted interval (<10% $CaCO_3$) with common siderite and calcite nodules, suggesting a migration of carbonate within clayey series. An interval with similar characteristics (clay-rich strata with common carbonate concretions, traces of siderite) is present in the *obtusum/oxynotum* zone transition in

the Mochras borehole (Woodland et al., 1971). The significant correlation (r = 0.68, n = 96, p-value < 0,05) between the carbonate content and the $\delta^{13}C_{carb}$ values could reflect the disturbance of the inorganic carbon signal in a clayey interval more



sensitive to diagenesis. A disturbance of the isotopic signal of carbon and oxygen from carbonates in this more clayey interval is likely especially when measurements taken on the carbon of organic matter and macrofossils do not show parallel variations (Ullmann unpublished data).

Long distance correlations can be made based on the comparison of $\delta^{13}C_{org}$ fluctuations through the Lower Jurassic in several sedimentary basins, likely reflecting a primary signal (Storm et al., 2020), even if diagenetic impacts on organic compounds may locally be significant (Meyers, 1994; Lehmann et al., 2002). Consequently, in the case of the Mochras and Montcornet boreholes, the $\delta^{13}C_{org}$ seems to be much more reliable to constrain carbon-cycle perturbations during the Late Sinemurian, while $\delta^{13}C_{carb}$ and $\delta^{18}O$ values cannot be interpreted confidently in terms of environmental and climatic fluctuations.

**5.2 Environmental significance of clay mineral assemblages**

**5.2.1 Detrital sources of clay minerals**

Although clay minerals may be transported over long distances, the Welsh High and the Irish Massif were likely the main detrital sources of the Cardigan Bay Basin (Dobson and Whittington, 1987; Xu et al. 2018). In the coeval deposits of the Dorset coast, southern England, the clay assemblages show a similar composition, but smectite is less abundant, suggesting that

detrital inputs into the Wessex Basin originated from distinct detrital sources including the Cornubian and the Armorican massifs (Schöllhorn et al., 2020a). In the Paris Basin, the main detrital sources of clay minerals were probably the proximal Palaeozoic massifs, including the London-Brabant Massif (LBM), the Armorican Massif and the Massif Central (Muller et al., 1973; Debrabant et al., 1992; Thierry et al., 2000), as suggested by dominant continental organic matter preserved in the Jurassic sediments drilled at Montcornet (Disnar et al., 1996).

The abundance of illite and chlorite reflects the intensity of erosion of these continental areas. Precambrian and/or Palaeozoic mudrocks from the Welsh High and from the LBM mainly contain mica-illite, chlorite, and corrensite (regular chlorite/smectite mixed-layer) reflecting deep burial and low-grade metamorphism (Lefrançois et al., 1993; Han et al., 2000; Merriman, 2006; Hillier et al., 2006). These assemblages occur in most Variscan massifs in Europe, and as a result, the high proportions of chlorite and illite in most Jurassic sedimentary successions of northwest Europe reflects mostly the erosion of these Palaeozoic

and older rocks (Jeans et al., 2001). In the Mochras borehole, from the base to the top of the Upper Sinemurian, the proportions of illite increase from ca. 20 to 35%. This evolution suggests that during the Late Sinemurian, an increasing erosion of the Welsh High basement occurred, compared to the development of thick soils. This may be the consequence of a long-term uplift of the Welsh High aided by faulting. By comparison, the constant proportions of illite recorded in the Montcornet borehole, located proximately south of the LBM, suggest continuous unchanged processes of erosion on this tectonically stable

massif.

In sediments, smectites have various origins. These minerals are most often reworked from soils where they formed under warm and seasonally humid climate (Chamley, 1989). But smectite can also be formed in marine environments either at the expense of volcanic glass or as an authigenic phase in slowly deposited sediment (Deconinck and Chamley, 1995). The



Sinemurian strata from the Mochras borehole do not show any evidence of volcanic origin and were rapidly deposited, with
high sedimentation rates responsible for their particularly high thickness (220 m for the Upper Sinemurian). The duration of
the Upper Sinemurian is estimated at 3 Myr, which suggests an average sedimentation rate (after compaction) of more than
70 m/Myr (Storm et al., 2020). In the *macdonnelli-aplanatum* subzones, a high mean sedimentation rate of ca. 40 m/Myr can
be estimated considering a duration of about 800 kyr of these subzones, according to cyclostratigraphical studies (Ruhl et al.,
2016). These relatively high sedimentation rates are not favourable to smectite authigenesis and consequently, most smectite
minerals identified here are likely detrital and originated from pedogenic blankets developed over the Welsh or Irish massifs
during periods of warm and seasonally humid climate.

Kaolinite, as for illite or chlorite, can be reworked from kaolinite-bearing sedimentary rocks and from the palaeoweathering
profile in continental areas (Hurst, 1985). Kaolinite may be reworked from sandstones where its authigenic formation is
common as pore filling booklets, such as Devonian (Old Red Sandstone) and Carboniferous sandstones from southern England
and Wales (Hillier et al., 2006; Shaw, 2006; Spears, 2006). Kaolinite may also originate from soils formed under hot and
regularly humid climate (Chamley, 1989; Ruffell et al., 2002). In both cases, increasing proportions of kaolinite in a
sedimentary succession suggest enhanced runoff favoring erosional processes and/or hydrolysing climate.

### 5.2.2 Environmental control of the clay sedimentation

In the Mochras borehole, the clay mineralogy of the Upper Sinemurian is relatively similar to that observed in the overlying
Pliensbachian formations, where an antagonistic evolution in the relative proportions of smectite and kaolinite is recorded.
The alternation of kaolinite-rich and smectite-rich intervals was interpreted for the Pliensbachian to be the result of climate
fluctuations, respectively dominated by regularly humid periods and semi-arid conditions (Deconinck et al., 2019). This
climate mode seems to be established at least at the beginning of the Late Sinemurian on the Welsh High and the surrounding
massifs. In the northern Paris Basin, in the Montcornet borehole, such climatic fluctuations are not recorded, even though this
basin was located at a comparable palaeolatitude as the Cardigan Bay Basin. Apart from the smectite-rich interval occurring
at the transition between the Lower and the Upper Sinemurian and at the base of the *obtusum* Zone, the clay mineralogy is
rather uniform. In this borehole, such smectite-rich intervals referred as "smectite events" were also recorded in the
Pliensbachian sediments (Bougeault et al., 2017). The occurrence of these "smectite events" was interpreted as the result of
the lowering of the sea-level favouring the formation of smectite in soils developed on newly exposed lands of the LBM. The
Lower/Upper Sinemurian boundary and the base of the *obtusum* Zone are precisely characterised by relative sea-level
lowstands in the northwest European domain (Jacquin et al., 1998; Hesselbo 2008; Haq, 2018). Therefore, the smectite-rich
interval occurring in the Sinemurian of Montcornet is also interpreted as a consequence of the lowering of the sea level allowing
the formation of smectite on newly exposed lands of the LBM with its comparatively subdued relief. By contrast, during
relative sea-level highstand, the LBM was probably at least partly flooded, suggesting that this massif was already deeply
eroded and relatively flat as early as the Sinemurian. A similar behaviour of this massif regarding sea-level fluctuations lasted
until the Late Jurassic (Hesselbo et al., 2009). Consequently, the LBM being often submerged, it is probable that the clay




minerals deposited to the south of this massif had partly a more distant origin, and this may explain the difference with the
Cardigan Bay Basin. The sea-level highstand during the *oxynotum* Zone is likely responsible for the starvation of the Paris
Basin, with a reduced sedimentation rate and even the significant sedimentary gap equally observed on the Dorset coast (cf.
Hallam 1999, 2000; Coe and Hesselbo 2000; Hesselbo et al., 2020).

**5.2.3 Kaolinite-rich intervals- Mochras borehole**

The three kaolinite-rich intervals observed in the Mochras borehole are highlighted by the kaolinite/illite and smectite/kaolinite
ratios (respectively K/I and Sm/K Fig.10). These intervals occur: (1) at the base of the *obtusum* Zone (interval K1), (2) around
the boundary between the *obtusum* and *oxynotum* zones (interval K2), and (3) in the *raricostatum* Zone (interval K3). The
onset of a fourth kaolinite-rich interval is present at the Sinemurian/Pliensbachian boundary and lasted through to the lower
part of the *jamesoni* Zone (Deconinck et al., 2019).

These intervals may be associated with more humid conditions, but in detail, they occur in different tectonic and eustatic
settings. K1 occurs during a period of lowstand of the sea-level (Hesselbo, 2008). These conditions may have enhanced the
proportions of kaolinite since this mineral is well-known to be deposited preferentially in proximal environments due to
differential settling processes of clay minerals (Gibbs, 1977; Godet et al., 2008). Therefore, we suggest that the high
proportions of kaolinite could be the result of combined influences of a more humid climate and relative low sea-level
conditions. By contrast, K2 occurs during a period of a high sea level, favourable to the deposition of more smectite.
Consequently, as this kaolinite-rich interval is also the most prominent, it is probable that the climatic conditions were
particularly hydrolysing (wet and/or warm) from the upper part of the *obtusum* zone to the lower part of the *oxynotum* Zone.
The resulting significant detrital fluxes were probably responsible for a dilution of the carbonates, leading to a more clay-rich
sedimentation during this interval. The proportions of kaolinite in K3 are lower than in K1 and K2 and this interval is also
marked by the relative abundance of illite, suggesting more efficient erosion of the basement that may be linked to tectonic
influences. It is therefore possible that in K3, kaolinite may be partly reworked together with illite from the unmetamorphosed
rocks of the basement.
Interestingly, the three kaolinite-rich intervals seem to coincide with higher values of [87]Sr/[86]Sr ratio consistent with increasing
detrital influences linked to the acceleration of the hydrological cycle (Fig. 10). The most prominent increase of [87]Sr/[86]Sr ratio
precisely coincides with the most prominent increase of kaolinite (K2) that occurs around the transition between the *obtusum*
and the *oxynotum* Zones. It should be noted, however, that the fluctuations in [87]Sr/[86]Sr ratio correspond to a global signal while
the clay minerals register a local signal.
To summarise, the three kaolinite-rich intervals are indicative of increasing moisture. Of these, K2 occurring during the
uppermost part of the *obtusum* Zone and the lower part of the *oxynotum* Zone is of particular interest, as it is at least indicative
of a significant acceleration of the hydrological cycle. This interval is also characterised by low $\delta^{18}O$ values consistent with
warm conditions, but these low values may also reflect enhanced freshwater inputs as a consequence of increasing runoff (Fig.
7; Deconinck et al., 2019; Schöllhorn et al., 2020a;). In addition, these $\delta^{18}O$ values are likely to have been affected by diagenetic



influences (section V.1.2) and therefore cannot be held here as a reliable proxy of seawater temperature. In the Copper Hill
       borehole drilled close to Ancaster in Lincolnshire, East England this interval is characterised by the abundance of *Classopollis*
       also indicating warm climate and by a negative excursion of -2/-3 ‰ of $\delta^{13}C_{org}$ (Riding et al., 2013). Surprisingly, this negative
       excursion is less clearly recorded in organic carbon at Mochras (Fig.10; van de Schootbrugge et al., 2008; Storm et al., 2020).

**5.3 Carbon cycle evolution and SBPE record**

A net decrease of the $\delta^{13}C_{org}$ values initiated from the late *oxynotum* Zone or the early *raricostatum* Zone and culminating in
       a marked negative excursion of the $\delta^{13}C_{org}$ near the Sinemurian/Pliensbachian boundary is clearly observed in the two sites.
       The amplitude of the decrease reaches -1‰ (-25 to -26‰) in the Paris Basin and -3‰ (-25 to -28‰) in the Cardigan Bay
       Basin, confirming the negative shift, highlighted by Storm et al. (2020). The condensation or even a significant hiatus in the
       northern part of the Paris Basin at the Sinemurian/Pliensbachian partly shortens this excursion and is therefore likely
responsible for the difference in the amplitude of the negative excursion recorded in the two sites (Bougeault et al., 2017;
       Mercuzot et al., 2019). This decrease is equally recorded in the Sancerre Borehole in the southern Paris Basin (Fig. 1) from
       the early *raricostatum* Zone and shows an amplitude of -4 ‰ (Peti et al., 2017). The environmental significance of this
       pronounced shift can be addressed as it seems to correspond to a longer episode than the SPBE centered on the boundary or
       even beginning in the early Pliensbachian (*jamesoni* Zone) and estimated with a duration of 2 Myr on the basis of
cyclostratigraphical analyses performed on the Mochras core (Ruhl et al., 2016). The stratigraphical evolution of the $\delta^{13}C_{org}$
       could be due to changes in the type of organic matter (Suan et al., 2015). However, no change is observed in the type of organic
       matter of the Upper Sinemurian deposits of Mochras according to van de Schootbrugge et al. (2005) that highlight a dominant
       continental type. Similar conclusions arise from the study of organic matter preserved in the Montcornet borehole (Disnar et
       al., 1996). As this organic carbon trend is well recorded both in the bulk organic matter ($\delta^{13}C_{org}$) resulting of a mixture of
continental and marine components and in the macrofossil wood (Storm et al., 2020), the best explanation would be a
       progressive and long term input of $^{12}C$ in both atmospheric and ocean reservoirs, that could be triggered by the volcanic activity
       and hydrothermalism linked to the CAMP, notably through the opening of the Hispanic Corridor (Price et al., 2016, Ruhl et
       al., 2016).
       While this negative carbon excursion from the *raricostatum* Zone is well recorded in the $\delta^{13}C_{carb}$ signal in the Montcornet
borehole, it is surprisingly not visible in the carbonate record at Mochras, where a negative shift is recorded later at the
       beginning of the Pliensbachian and recognised as corresponding to the Sinemurian Pliensbachian Boundary Event (SPBE,
       Ruhl et al., 2016). The discrepancy between the two isotopic signals in the Mochras Borehole could be due to local diagenetic
       effect that overprint the primary signal of carbonates, as is probably also the case for the *oxynotum* Zone. Thus, the onset of
       the negative trend at the Sinemurian/Pliensbachian transition is from the early *raricostatum* Zone, while the shift reaches a
maximum during the *jamesoni* Zone (plateau of low $\delta^{13}C_{org}$ values, Storm et al., 2020) resulting from a long and progressive
       increase of light carbon release in relation with volcanism activity. Some authors (Mercuzot et al., 2019; Schöllhorn et al.,



2020a, b) previously mentioned such differences in the onset and record of the SPBE recognised in various basins worldwide which may express local diagenetic and environmental effects. This may explain why the runoff conditions, supported in this study by the kaolinite content, are firstly decreasing (i.e. *raricostatum* Zone) before drastically increasing during the *jamesoni*
Zone (Deconinck et al., 2019) which should correspond to the maximal disturbance in the carbon cycle concomitant to warmer temperatures and enhanced precipitation.

## 6 Conclusions

The study of the clay mineralogy at a high resolution in the Upper Sinemurian of the Mochras and Montcornet boreholes shows that the clay minerals are mainly detrital and come from the erosion of the basement and the soil cover of the Palaeozoic
massifs. The thick and continuous succession penetrated by the Mochras borehole shows significant fluctuations in the relative proportions of clay minerals. The stratigraphical succession shows an inverse relationship between the proportions of kaolinite and smectite, which probably results from an alternation of warm and humid periods with semi-arid periods. This climatic mode previously identified in the overlying Pliensbachian therefore seems to be in place at least from the Late Sinemurian. The end of the *obtusum* Zone and the *oxynotum* Zone correspond to a particularly hot and humid period favourable to a strong
runoff at the origin of significant terrigenous inputs probably responsible for a repression of carbonate sedimentation. This particular hot and humid interval is also expressed by the abundances of *Classopollis* and *Liasidium variabile*, by a slight negative excursion of $\delta^{13}C_{org}$ and by low values of $\delta^{18}O$. However, although the low values of $\delta^{18}O$ are consistent with high temperatures, they may also result from a decreasing salinity of seawater due to increased supplies of fresh water and/or a likely local diagenetic influence.
In the Montcornet borehole, to the north of the Paris Basin, the thinner succession has many hiatuses, notably that of the *oxynotum* Zone. The clay minerals are similar to those identified in the Cardigan Bay Basin, but the discontinuous series does not allow the alternation of humid and semi-arid periods to be identified. It is possible that this also results from the different and more distant origin of clays. In this borehole, a smectite-rich interval is identified within the *obtusum* Zone during a period of lowstand of sea level. This smectite interval, like those identified in the overlying Pliensbachian, would result from the
emergence of the London-Brabant Massif then subjected to active pedogenesis. A eustatic control of the clay sedimentation is therefore expressed along the London-Brabant Massif, a situation previously proposed for the Pliensbachian, and which lasted until the end of the Jurassic.

Unlike the clay diagenesis, which is negligible, carbonate diagenesis notably expressed as nodulisation causes a dispersion of the isotopic values of $\delta^{13}C$ carb, as well as $\delta^{18}O$ carb whose interpretation in terms of palaeotemperature is unsound. The
evolution of $\delta^{13}C_{org}$ reveals a progressive decrease during the *raricostatum* Zone before the very low values at the Sinemurian/Pliensbachian transition corresponding to the SPBE.



**Acknowledgments:** The authors warmly thank J.L. Dommergues who determined ammonites newly collected during the description and sampling of the Montcornet borehole. We also thank Claude Aurière (Andra) for providing the cores from the A901 borehole. S.P.H and C.V.U. acknowledge funding from the UK Natural Environment Research Council (NE/N018508/1). J.B.R. publishes with the approval of the Chief Executive Officer, British Geological Survey (NERC). This is a contribution to the JET (Jurassic Earth Time) Project.

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

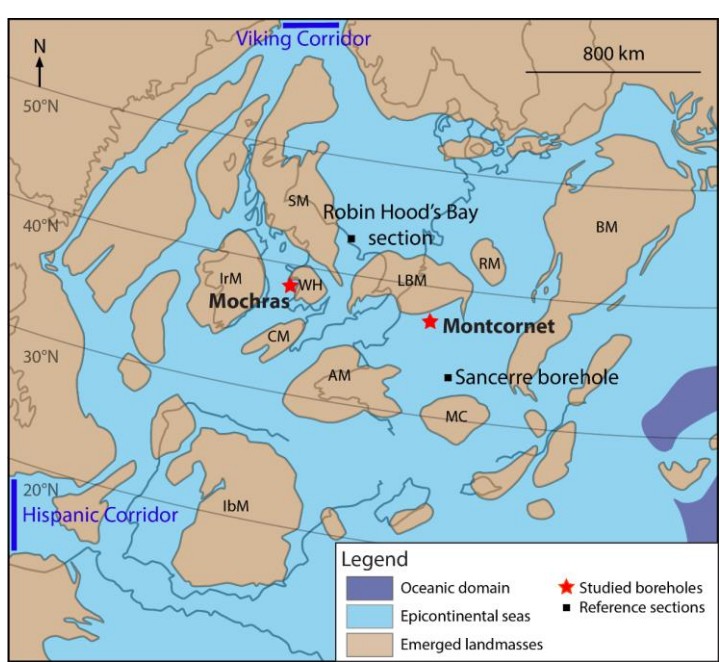

**Figure 1: Upper Sinemurian palaeogeographical map of the northwest Tethyan domain and position of the studied boreholes
(modified from Bougeault et al., 2017). Abbreviations: SM = Scottish Massif; IrM = Irish Massif; CM = Cornubian Massif; WH = Welsh High; LBM = London-Brabant Massif; RM = Rhenish Massif; BM = Bohemian Massif; MC = Massif Central; AM = Armorican Massif; IbM = Iberian Massif.**



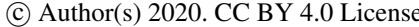

**Figure 2: Location map of the Mochras borehole, simplified lithology and biostratigraphy of the Upper Sinemurian (from Copestake and Johnson, 2014). Abbreviations: L.Sin. = Lower Sinemurian, Pli. = Pliensbachian,** *turn.* **=** *turneri***.**



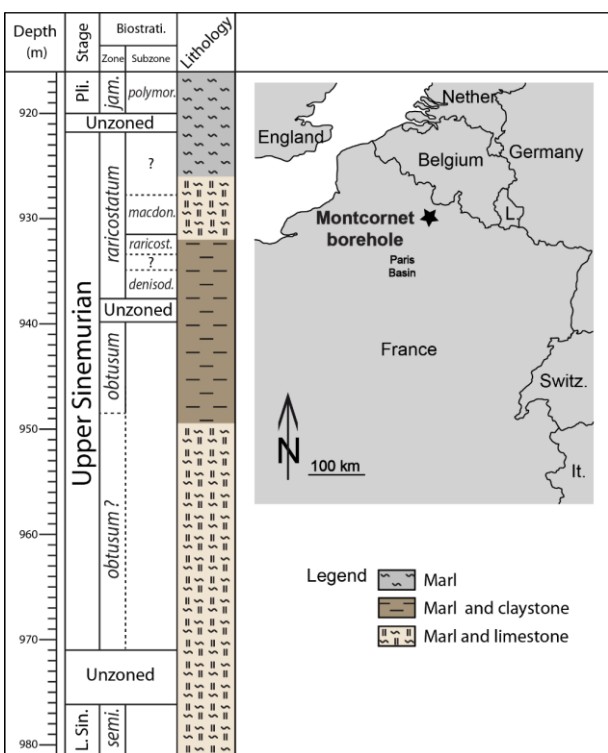

**Figure 3: Location map of Montcornet borehole, simplified lithology and biostratigraphy of Upper Sinemurian (from Fauconnier, 1995; Yang et al., 1996).** *densinod.* = *densinodulum.*, *jam.* = *jamesoni*, L.Sin = Lower Sinemurian, *macdonn.* = *macdonnelli*, Pli. = **Pliensbachian,** *polymor.* = *polymorphus*, *raricost.* = *raricostatum*, *semi.* = *semicostatum.*




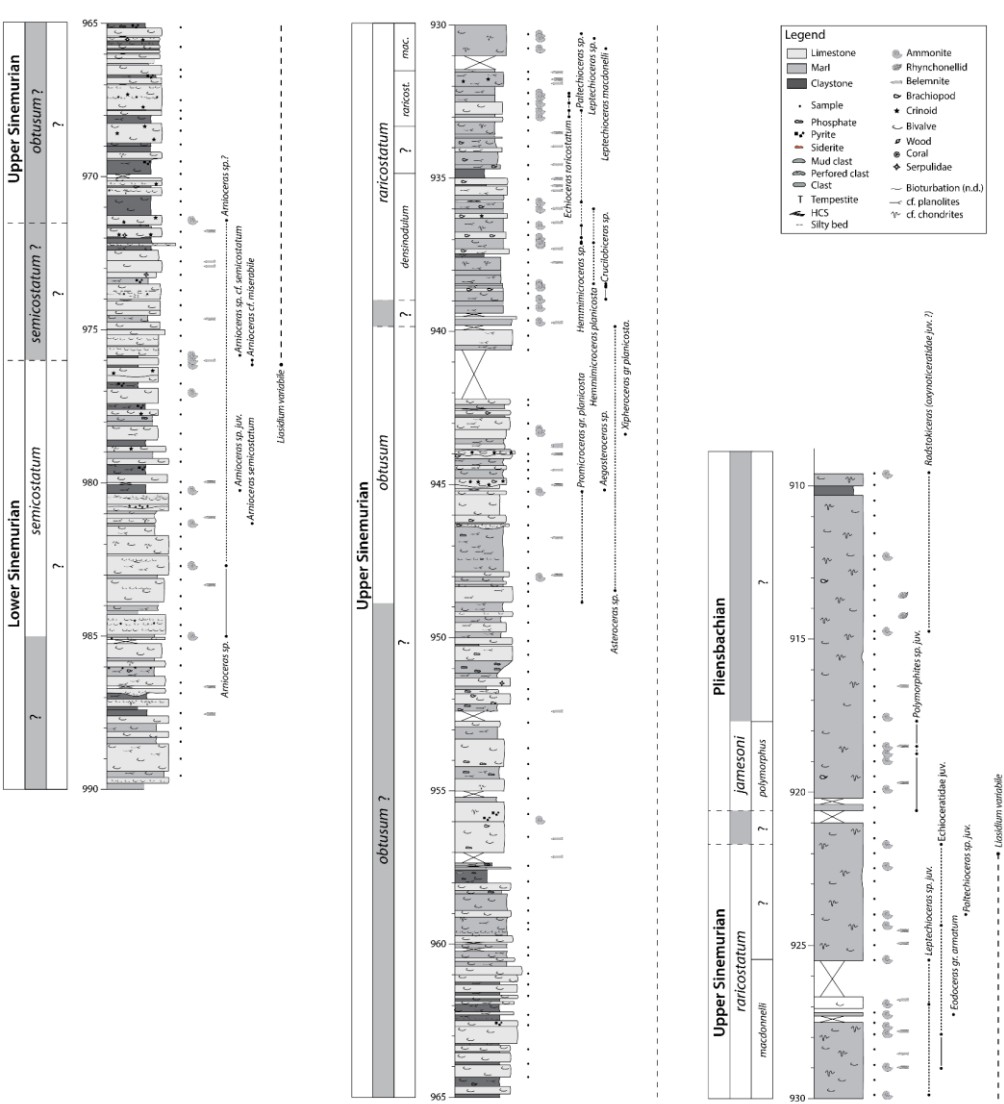

**Figure 4: Detailed lithology, sampling and biostratigraphy of Upper Sinemurian of the Montcornet borehole. Biostratigraphy modified from Yang et al. (1996) and Fauconnier (1995). Gr. – Group, juv. – Juvenile, *mac.* = *macdonnelli*, *semico.* = *semicostatum*.**






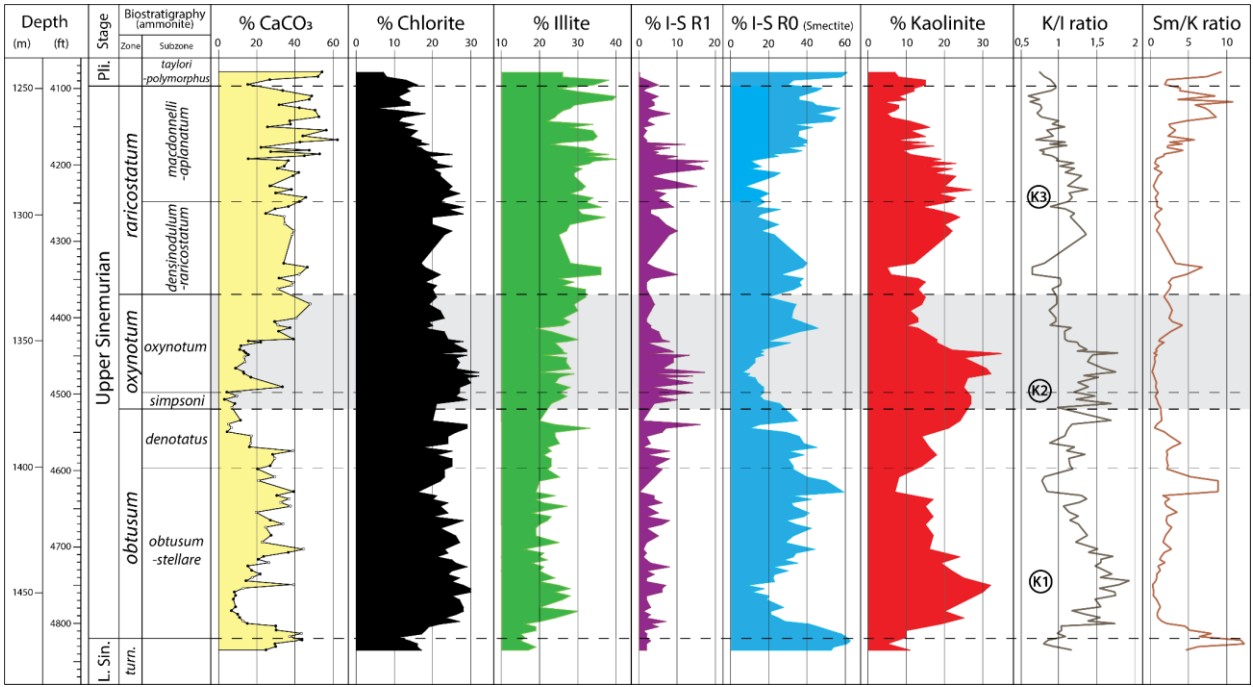

**Figure 5: Calcite proportions and composition of the clay fraction of Upper Sinemurian strata of the Mochras borehole.**
**Kaolinite/illite (K/I) and smectite/kaolinite (Sm/K) ratios corresponds to the ratio of the areas of the main peaks of these minerals.**
**L.Sin. – Lower Sinemurian, Pli. – Pliensbachian,** *turn. = turneri.*





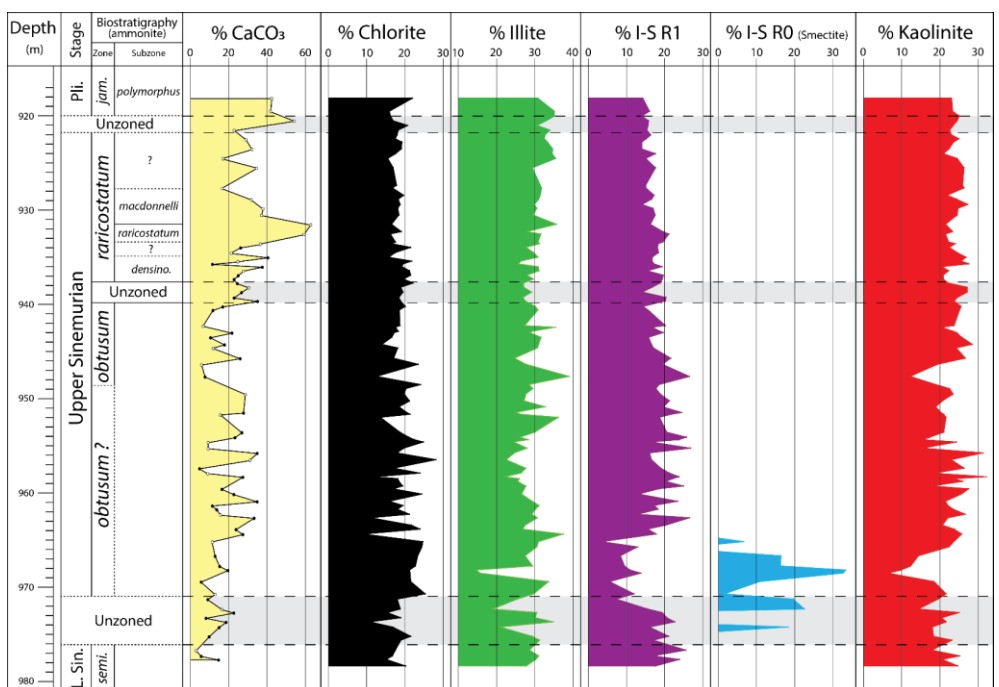

**Figure 6: Calcite proportions and composition of the clay fraction of Upper Sinemurian strata of the Montcornet borehole.** *densinod.*
*= densinodulum, macdonn. = macdonnelli,* **Pli.** *= Pliensbachian, polymor. = polymorphus, raricost. = raricostatum.*

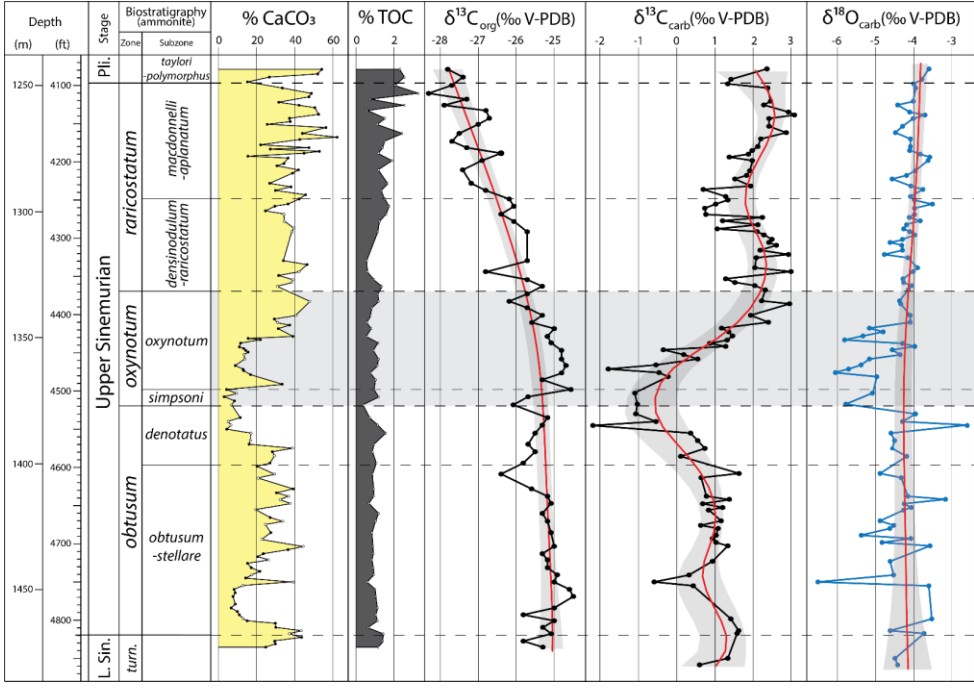





**Figure 7: Carbon (δ¹³C$_{carb}$ and δ¹³C$_{org}$) and oxygen (δ¹⁸O$_{carb}$) isotopes of the Upper Sinemurian on the Mochras borehole coupled with the proportions of CaCO₃ and TOC. Smoothing curve in red and 95% confidence interval in grey (Kernel regression). L.Sin. = Lower Sinemurian, Pli. = Pliensbachian, *turn.* = *turneri*.**


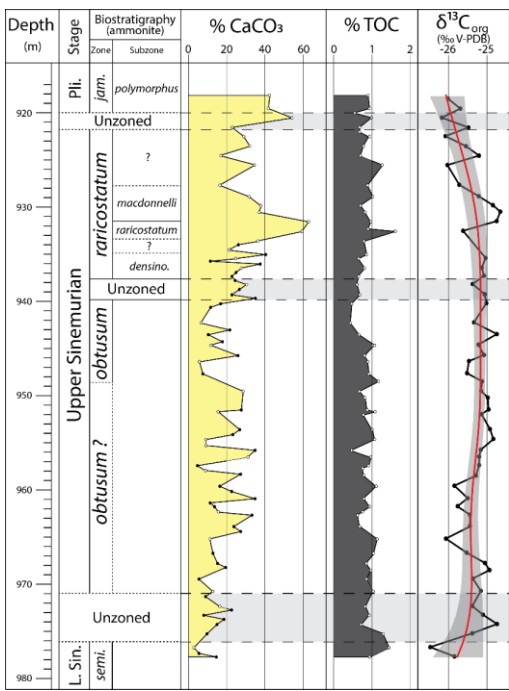

**Figure 8: Carbon isotopic ratio (δ¹³C$_{org}$) of Upper Sinemurian strata of the Montcornet borehole coupled to CaCO₃ and TOC proportions. Smoothing curve in red and 95% confidence interval in grey (Kernel regression). *densinod. = densinodulum*, L.Sin. = Lower Sinemurian, *macdonn. – macdonnelli*, Pli. = Pliensbachian, *polymor. = polymorphus*, *raricost. = raricostatum*, *semi. = semicostatum*.**






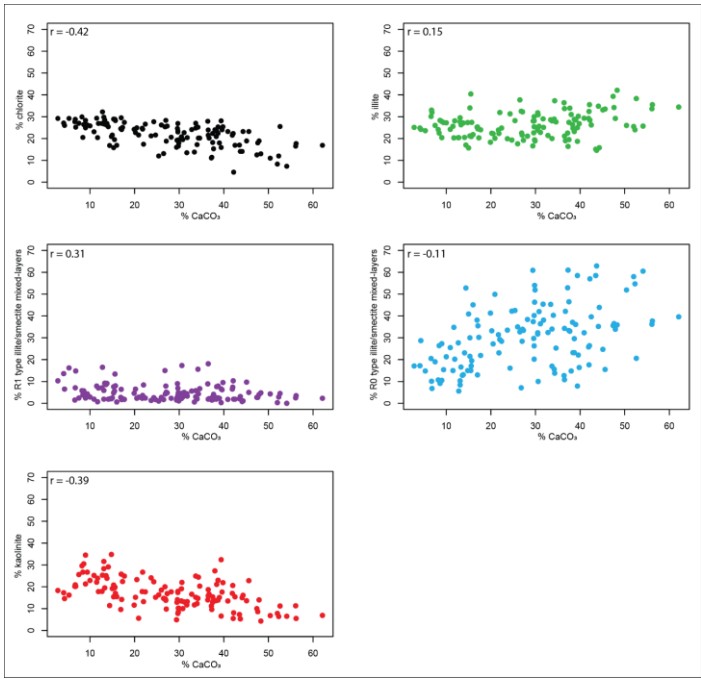

**Figure 9:** **Correlations between CaCO₃ content and the clay mineral relative proportions in the clay fraction in the Mochras**
**borehole.**





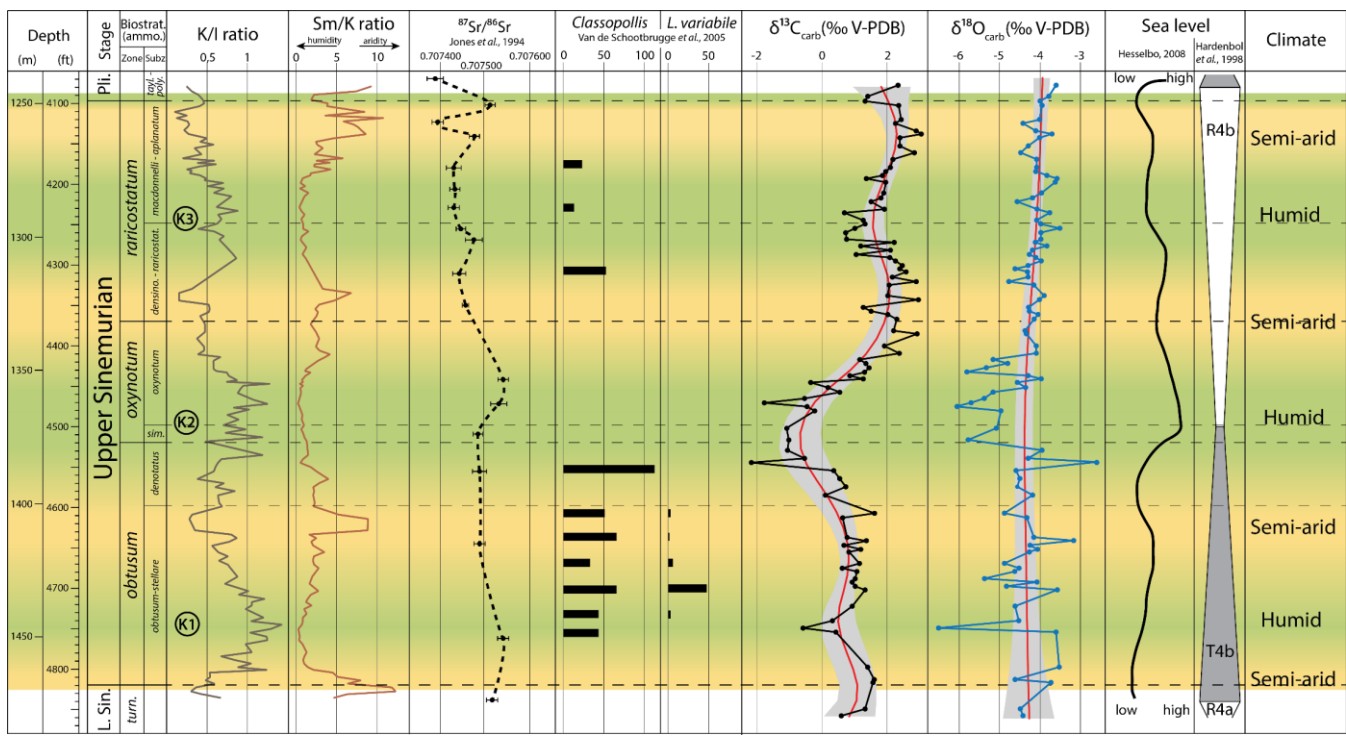

**Figure 10: Palaeoclimatic interpretations inferred from clay mineral assemblages expressed by K/I and Sm/K ratios, compared with** *Classopollis*, *Liasidium variabile* **abundance and strontium isotopes (Jones et al., 1994; van de Schootbrugge et al., 2005; Riding et** 805 **al., 2013), carbon and oxygen isotopes variations. Eustatic variations from Hardenbol et al. (1998) and Hesselbo (2008).**