# Peer review of "Million-year-scale alternation of warm-humid and semi-arid periods as a mid-latitude climate mode in the Early Jurassic (Late Sinemurian, Laurasian Seaway)"

_Climate of the Past, 2020_

## Short Comment (SC1) · 20 Oct 2020

please consider adding reference to the following paper:

Franceschi, M., Dal Corso, J., Cobianchi, M., Roghi, G., Penasa, L., Picotti, V., Preto, N., 2019. Tethyan carbonate platform transformations during the Early Jurassic (Sinemurian-Pliensbachian, Southern Alps); comparison with the Late Triassic Carnian Pluvial Episode. GSA Bulletin 131 (7/8). https://doi.org/10.1130/B31765.1.

this paper reports extensive d13Ccarb record and also d13Corg data calibrated with

nannoplankton biostratigraphy and ammonoid occurrences across the Sinemurian-Pliensbachian interval from the Lombardian Basin and Trento Platform and highlights the SPBE and also a negative d13C shift that is correlated to that identified by Masetti et al. (2017) and corresponding to the "Liasidium Event" of Riding et al. (2013) and Hesselbo et al. (2020)

———————————————————

---

## Referee Comment (RC1) · Anonymous Referee #1 · 1 Feb 2021

**Reviewer #1**

**Review of the Manuscript Number: cp-2020-99**

Title: Million-year-scale alternation of warm-humid and semi-arid periods as a mid-latitude climate  mode in the Early Jurassic (Late Sinemurian, Laurasian Seaway)

Author(s): Thomas Munier et al.

Article Type: Research Paper

**General comments**

The research is original, novel and considered as important to the field, so it is a good candidate to be published in CP.

The structure is appropriate and, in my opinion, the used language is correct.  The manuscript presents a substantial contribution to scientific progress within the scope of Climate of the Past.

The scientific approach and applied method referring the clay minerals are valid, but some of the isotopic data are not fully reliable and  they should not be used for palaeoclimatic interpretations. The results are discussed in an appropriated way and the references are appropriated.

The scientific results and conclusions are presented in a concise, clear and well-structured way, and the number and quality of figures is correct.

There are no major points of conflict, as it is a high quality palaeoclimatic study mainly based on the study of clay minerals reflecting an alternation of humid and semi-arid periods during the Late Sinemurian, comparing the data obtained in two boreholes drilled in western UK (Mochrasa) and northern France (Montcornet). However, the isotopic data, mainly obtained from the Mochras borehole, are strongly suspicious to be st4rongly affected by burial diagenesis, as the δ18O presented values are too low to reflect normal seawater values and cannot be used for palaeoclimatic studies.

Consequently, the paper would need a **MINOR REVISION**.

**Specific comments.**

The manuscript represents an analysis of the vertical distribution of clay mineral and stable isotope during the Late Sinemurian and the Sinemurian— Pliensbachian boundary, based on sampling of two boreholes drilled in the UK and in France.

Line 96. This latitude is also corroborated by the palaeomagnetic data presented by Osete et al., 2011 (Tectonophysics, 502, 105-120).

Line 216.Reader has to wait until line 216 to confirm that the drill holes have recovered a supposedly continuous core of the drilled sections. That should be specified before in the text, including the core diameter and percentage of recovery of the core. Could some of the gaps found in the Montcornet hole due to the loses of core in some intervals? I assume that the hole was drilled using the wireline method, but it would be convenient to specify that in the manuscript. If the drilled section is dipping, were the thicknesses corrected respect to the depths?

Lines 129 to 146. It is quite singular to perform ammonite biostratigraphy in cores, due to their limited diameter, especially in the case of the Montcornet hole, were as said in line 129-130, some important hiatuses occur, and the ammonites are relatively scarce. This does not support the "High resolution data " mentioned at the beginning of the Abstract.

Section 4.3.1. It seems clear that the isotopic data of the study are the weakest part of the paper.  Values of δ18O up to -6.54‰ reflect the presence of strong diagenetic overprints. Also the δ13C carb curve is completely different respect to the δ13C org curve, confirming the presence of the strong diagenetic overprints. As a consequence, none of the isotopic data are useful in a palaeoclimatic study (see line 324) as it is supposed to be the present paper. A diagenetic study of the carbonates is essential to be sure that your isotopic data reflect the original Jurassic seawater conditions. Why this diagenetic study has not been performed?

Line 260. "Abundant smectite indicate a limited diagenetic influence". In the paragraph above it seems that the diagenesis in the carbonates of the Mochras borehole is not negligible, but is it in the clays? Some additional justifications together with the shallow depth of burial would be welcome.

Section 5.1.2. Even of δ18O isotopic data from the Montcornet borehole are not included, it seems that diagenetic overprints are also present, showing numerous indications in the lithology. However the δ13C org curves show similar trends in both localities, indicating that this data could be reliable.

The main concern is to be sure that the climatic fluctuations are not the result or are influenced by the diagenetic processes.

Line 399. The ammonites zone or Zone should be uniform. Better obtusum Zone. Please check the rest of the text.

Line 407-408. Obtusum and the oxynotum zones.

Line 412. "Low δ18O values consistent with warm conditions". In previous sections it has been established that δ18O values cannot be used as a palaeoclimatic criteria, so it should not be

used here as indicative of warm conditions, and this is contradictory with the stated in the following lines of the manuscript.

Lines 416-417. Reference(s) supporting the interpretation of *Classopollis* as an indicator of warm clmate is needed*. Clasopollis* is a long-term pollen showing  a distribution probably from the Triassic up to the beginning of the Paleogene (Vakhrameyev, 1980) surviving lots of climatic changes. So taking it as a good indicator of warming could be at least very risky, if it is not supported by more reliable data.

Line 417-418. "Surprisingly this negative $\delta 13C$ org excursion is less clearly recorded in inorganic carbon at Mochras than in the Copper Hill drillhole". This could be another indication of the strong diagenetic overprint at Mochras.  That could be an indication that the multiple papers based in the isotopic signal of the Mochras borehole are values affected by the diagenetic overprint, reflecting local conditions and no global ones.

Line 425-438. It would be a nice explanation but, has been compared the age of the SPBE $\delta 13C$ sift with the absolute ages of the CAMP emissions? This data should be incorporated into the manuscript and supported with more data.

Line 461-462. "Hot and humid interval ….. expressed by  low values of $\delta 18O$". Again, this $\delta 18O$ values cannot be used for palaeoclimatic interpretations.

---

## Referee Comment (RC2) · Anonymous Referee #2 · 2 Mar 2021

This paper is an interesting contribution to the paleoclimatic reconstruction of the Early Jurassic using a multi proxy approach including Clay minerals and stable isotopes. This on 2 cores located Mochras in the Cardigan bay basin (Mochras borehole) and the Paris Basin Montcornet borehole). The topic fits therefore well with the scope of the CP journal.

The paper is well written and well structured. The figures are informative and of good quality. This Ms can only be accepted only after medium to major thorough revisions, since I have some important concerns about the quality of the data and some interpretations, which are not always supported by the data.

Sample resolution

In the first line of the abstract, the authors claims that it is a high resolution study (223 clay analyses). High-resolution is may be a slight overstatement, since (if we look at figures 5-6) only some 70-80 clay samples have been analysed along a 200m section at Mochras (1 sample/2.5m). The sample resolution is a little bit better in the Moncornet Borehole (around 60 samples for a 60m thick section.

Biostratigraphy

It looks that all the biostratigraphy is based on ammonites, it is maybe OK for the Mochras core, but not so evident for the Moncornet borehole, where several marquers are missing. It would be good to complete the biostratigraphy using nannofossils. At lines 128, the authors claim that the section is complicated by some important hiatuses and scarcity of ammonites. It would be important to discuss and especially locate these hiatuses. The upper Sinemurian are made of Gryphaea accumulations, probably resulting from storms interrupted by P- rich condensed levels. This makes the correlation quite difficult and some of the ammonites may be reworked.

Stable isotopes

This is the weakest part of this paper. $\delta$18O values are significantly too negative and reflects a strong diagenetic overprint. I agree that these sediments have not been too much buried, since smectite and kaolinite are still present. But it does not mean that other diagenetic processes were not acting. The presence of siderite is a good indication of a strong diagenetic process. It would have been good to analyse the bulk mineralogy by XRD (easy and fast to perform). Moreover, the most negative values of both $\delta$18O and $\delta$13Ccarb occur in levels, in which calcite contents are quite low (<15%). Some simple cathodoluminescence analyses would help to retrace the diagenetic story of these sediments. $\delta$18O and $\delta$13Ccarb can't be use for paleoclimatic reconstructions

Interactive
comment

as the authors did in their figure 10 or at line 30 of the abstract. This is clearly confirmed by the observed discrepancies between the $\delta$13Ccarb and $\delta$13Corg. At Mochras the $\delta$13Ccarb curve is really very different from the $\delta$13Corg. This must be discussed in details. The $\delta$13Ccarb shows a huge excursion in the oxynotum zone, which is not present in the $\delta$13Corg curve. The correlation between Mochras and Montcornet based on $\delta$13Corg curves is not convincing, since very are too many hiatuses. The authors must also explain why the $\delta$13Corg values are more negative in the raricostatum zone of the Mochras core (down to -28) compared with coeval Montcornet values (-26). This maybe due to a difference in organic matter origin (see Schoellhorn et , 2020 or Suan et al, 2015). In addition, the authors may try to correlate their $\delta$13Corg curve with the one published by Peti et al, 2016, which appears to show a different trend. I suggest also to examine the $\delta$13Corg published by Schoellhorn et al, 2020 (Dorset section), which shows several shifts in the upper Sinemurian, which can't be found neither at Mochras nor at Montcornet. Note also that Schoellhorn et al (2020,) found a negative shift in both $\delta$13Ccarb and $\delta$13Corg curves in the obtusum zone, confirming that the isotopic data from both Mochras and Montcornet cores are quite suspicious and can't really used for correlation. It would be good to try to correlate these isotopic records together.

Clay minerals

This is the most interesting part of this MS. The alternation of humid and semi-arid periods during the Late Sinemurian at Mochras is very convincing and their paleoclimatic interpretation is correct. However, it is not the case at Montcornet, where these cycles are not present. Contrary to Mochras, the kaolinite is not showing significant variations (20-30%). Since there is almost no smectite at Montcornet, I understand that the authors can't provide a SM/K ratio for that core, but they could have shown the K/I ratio, which exhibits at Mochras nice cycles showing that illite and potentially chlorite are not coupled with kaolinite, which may have originated from coeval paleosoils weathering. A different trend seems to characterize the clays distribution at

MontCornet, where kaolinite, illite and chlorite shows the same trend (a simple statistic multivariate approach would be very helpful). I am therefore not convinced that the two cores can be correlated based on clay minerals. At line 405, the authors underline the good correlation with the most prominent kaolinite increase with increased Sr ratio in the obtusum-oxynotum zones. Interestingly, this interval corresponds to very high CIA values (Schöllhorn et al,2020). The absence of smectite is difficult to understand and must be better explained. At line 465, the authors wrote that the different clay minerals trends may be due to the fact that Montcornet was located in a more distal location than Mochras. If it is the case, I would expect more smectite and it is really not the case. The authors linked the high amounts of smectite with sea-level low and the erosion of London-Brabant Massif. This is rather unlikely, since high smectite contents are generally linked with high sea-level (e.g. Godet et al, 2008, Ruffel et al, 2002, Gibbs et al, 1977). Moreover, sea-level lows are characterized by a mix of clay minerals such as illite, chlorite, kaolinite..etc (Deckoninck, 1985). I suggest that the authors try to correlate their clay minerals data with the ones published by Schöllhorn et al (2020) in the Dorset. The upper Sinemurian (even if more condensed) is characterized by similar K/I and Sm/K cycles confirming that these cycles can be globally correlated and represent true paleoclimatic (semi-arid-humid) changes.

---

## Author Comment (AC2) · 17 Apr 2021

General comments

This paper is an interesting contribution to the paleoclimatic reconstruction of the Early Jurassic using a multi proxy approach including Clay minerals and stable isotopes. This on 2 cores located Mochras in the Cardigan bay basin (Mochras borehole) and the Paris Basin Montcornet borehole). The topic fits therefore well with the scope of the CP journal. The paper is well written and well structured. The figures are informative

and of good quality. This Ms can only be accepted only after medium to major thorough revisions, since I have some important concerns about the quality of the data and some interpretations, which are not always supported by the data.

We thank the referee for assessing our work and for providing an important review.

Sample resolution

In the first line of the abstract, the authors claims that it is a high resolution study (223 clay analyses). High-resolution is may be a slight overstatement, since (if we look at figures 5-6) only some 70-80 clay samples have been analysed along a 200m section at Mochras (1 sample/2.5m). The sample resolution is a little bit better in the Montcornet Borehole (around 60 samples for a 60m thick section.

The term "high resolution" has been removed.

Biostratigraphy

It looks that all the biostratigraphy is based on ammonites, it is maybe OK for the Mochras core, but not so evident for the Montcornet borehole, where several marquers are missing. It would be good to complete the biostratigraphy using nannofossils.

Nannofossils biostratigraphy is not available in Montcornet borehole but the magne-tostratigraphy has been added (Yang et al., 1996 and Moreau et al 2002, this latter ref. has been added) for Montcornet (Fig.4), as it is currently used as a reference for the Sinemurian in the GTS 2020.

At lines 128, the authors claim that the section is complicated by some important hia-tuses and scarcity of ammonites. It would be important to discuss and especially locate these hiatuses.

We have modified the text dealing with the biostratigraphy based on ammonites of the Montcornet borehole.

The upper Sinemurian are made of Gryphaea accumulations, probably resulting from

storms interrupted by P- rich condensed levels. This makes the correlation quite diffi-cult and some of the ammonites may be reworked.

Gryphaea accumulations are common in the Lower Sinemurian succession and rarer in Upper Sinemurian. There is however no evidence of reworked ammonites. Effectively, scattered phosphate nodules have been observed, but no P-rich condensed levels are associated with Gryphaea accumulations that could be interpreted as condensed horizons.

Stable isotopes

This is the weakest part of this paper. $\delta$18O values are significantly too negative and reflects a strong diagenetic overprint. I agree that these sediments have not been too much buried, since smectite and kaolinite are still present. But it does not mean that other diagenetic processes were not acting. The presence of siderite is a good indication of a strong diagenetic process. It would have been good to analyse the bulk mineralogy by XRD (easy and fast to perform). Moreover, the most negative values of both $\delta$18O and $\delta$13Ccarb occur in levels, in which calcite contents are quite low (<15%). Some simple cathodoluminescence analyses would help to retrace the diagenetic story of these sediments.

Yes, we agree! $\delta$18O and $\delta$13Ccarb have been completely removed as paleoclimatic proxies on now fig.11 and text. So we modified the text consequently. In our opinion, the $\delta$18O values are shifted to low values and $\delta$13Ccarb values cannot be used as en-vironmental proxies because of carbonate diagenesis. Bulk mineralogy shows indeed the occurrence of siderite (nodules observed in the core) indicating that significant car-bonate diagenesis disturbed the original signal.

$\delta$18O and $\delta$13Ccarb can't be use for paleoclimatic reconstructions as the authors did in their figure 10 or at line 30 of the abstract. This is clearly confirmed by the observed discrepancies between the $\delta$13Ccarb and $\delta$13Corg.

$\delta$18O and $\delta$13Ccarb have been removed from now Figure 11 and their reference as a paleoclimatic marker has been removed from the abstract and conclusion.

At Mochras the $\delta$13Ccarb curve is really very different from the $\delta$13Corg. This must be discussed in details. The $\delta$13Ccarb shows a huge excursion in the oxynotum zone, which is not present in the $\delta$13Corg curve.

Difference between $\delta$13Ccarb and $\delta$13Corg curves seems to be related to the impact of diagenesis on carbonates. The $\delta$13Ccarb negative excursion of the oxynotum zone is likely the result of early diagenetic processes in this depleted carbonate interval (e.g. Ader & Javoy 1998). The $\delta$13Corg is probably more reliable as an environmental proxy since a similar evolution is recorded in several sedimentary basins as shown by a newly added figure (Fig. 10) of correlation that show a consistent $\delta$13Corg signal between UK and French basins .

The correlation between Mochras and Montcornet based on $\delta$13Corg curves is not convincing, since very are too many hiatuses. The authors must also explain why the $\delta$13Corg values are more negative in the raricostatum zone of the Mochras core (down to -28) compared with coeval Montcornet values (-26). This maybe due to a difference in organic matter origin (see Schoellhorn et al, 2020 or Suan et al, 2015).

Yes we totally agree on the role of hiatuses in the Montcornet borehole (oxynatum Zone and the upper part of the raricostatum Zone – aplanatum subzone). The new figure of correlation (fig. 10) highlights the role of these hiatuses. Taking into account these hiatuses we can see that the isotopic are similar between UK and France. New data from Storm et al. (2020) indicates a potential shift in organic matter origin that may exacerbate SPBE. This point is discussed now in the MS

In addition, the authors may try to correlate their 13Corg curve with the one published by Peti et al, 2016, which appears to show a different trend. I suggest also to examine the $\delta$13Corg published by Schoellhorn et al, 2020 (Dorset section), which shows several shifts in the upper Sinemurian, which can't be found neither at Mochras nor

at Montcornet. Note also that Schoellhorn et al (2020,) found a negative shift in both $\delta$13Ccarb and $\delta$13Corg curves in the obtusum zone, confirming that the isotopic data from both Mochras and Montcornet cores are quite suspicious and can't really used for correlation. It would be good to try to correlate these isotopic records together.

It was done with the new figure 10.

Clay minerals

This is the most interesting part of this MS. The alternation of humid and semi-arid periods during the Late Sinemurian at Mochras is very convincing and their paleoclimatic interpretation is correct. However, it is not the case at Montcornet, where these cycles are not present. Contrary to Mochras, the kaolinite is not showing significant variations (20-30%). Since there is almost no smectite at Montcornet, I understand that the authors can't provide a SM/K ratio for that core, but they could have shown the K/I ratio, which exhibits at Mochras nice cycles showing that illite and potentially chlorite are not coupled with kaolinite, which may have originated from coeval paleosoils weathering. A different trend seems to characterize the clays distribution at Montcornet, where kaolinite, illite and chlorite shows the same trend (a simple statistic multivariate approach would be very helpful).

Yes we agree.

I am therefore not convinced that the two cores can be correlated based on clay minerals.

Yes we agree, the two boreholes cannot be correlated using clay minerals as sources are likely different.

At line 405, the authors underline the good correlation with the most prominent kaolinite increase with increased Sr ratio in the obtusum-oxynotum zones. Interestingly, this interval corresponds to very high CIA values (Schöllhorn et al,2020).

Line 408, the relationship between CIA highlighted by Schöllhorn et al. (2020) and the

increase in kaolinite was added.

The absence of smectite is difficult to understand and must be better explained. At line 465, the authors wrote that the different clay minerals trends may be due to the fact that Montcornet was located in a more distal location than Mochras. If it is the case, I would expect more smectite and it is really not the case. The authors linked the high amounts of smectite with sea-level low and the erosion of London-Brabant Massif. This is rather unlikely, since high smectite contents are generaly linked with high sea-level (e.g. Godet et al, 2008, Ruffel et al, 2002, Gibbs et al, 1977). Moreover, sea-level lows are characterized by a mix of clay minerals such as illite, chlorite, kaolinite..etc (Deckoninck, 1985).

We do not agree with this comment. It is true that usually the proportions of smectites are more important during periods of high sea level (e.g. Deconinck and Chamley, 1995), partly due to the differential sedimentation of clays, but on the border of the London-Brabant massif, the situation is particular. In reality, in the Jurassic (but also in the Cretaceous), this very flattened massif was very often submerged (contrary to what is indicated on most paleogeographic maps) and consequently, the clay sedimentation on its borders was the result of more distant contributions. However, during periods of low sea level, this massif had emerged and smectite pedogenesis could develop. It is clear that this massif constitutes the source of smectite. This very particular situation was highlighted in the Kimmeridgian and the Tithonian of the North-West of the Paris Basin (Boulonnais) where the lower offshore facies are rich in illite and kaolinite and devoid of smectite, while the shoreface facies are rich in smectite (see e.g., Hesselbo et al 2009). This situation is identical in the Callovian/Oxfordian on the Ardennes border (Pellenard & Deconinck, 2006) as well as in the Pliensbachien (Bougeault et al., 2017), a publication in which we explain this singularity in detail.

I suggest that the authors try to correlate their clay minerals data with the ones published by Schöllhorn et al (2020) in the Dorset. The upper Sinemurian (even if more condensed) is characterized by similar K/I and Sm/K cycles confirming that these

cycles can be globally correlated and represent true paleoclimatic (semi-arid-humid) changes.

Yes, we agree, but the very different resolution of Iris Schollhorn's study makes the comparison quite difficult. However, we added a sentence in the text indicating that the results presented in Schollhorn et al 2020 are quite comparable with ours.

Please also note the supplement to this comment:
https://cp.copernicus.org/preprints/cp-2020-99/cp-2020-99-AC2-supplement.pdf

———————————————————————

[Figure]

**Fig. 1.**

[Figure]

**Fig. 2.**

[Figure]

**Fig. 3.**

---

## Author Comment (AC3) · 17 Apr 2021

Dear Matteo Franceschi, thank you for your comment, the reference you suggest is relevant and has been added to the revised manuscript.

---

## Author Response (AR1)

**Authors' response to reviewers' comments**

**Million-year-scale alternation of warm-humid and semi-arid periods as a mid-latitude climate mode in the Early Jurassic (Late Sinemurian, Laurasian Seaway)**

**(cp-2020-99)**

We thank the reviewers for accepting to review our manuscript and for their remarks which allowed us to improve our work. Authors' responses are written in blue and action on manuscript are in red.

**1) Summary of changes**

Changes are highlighted in the tracked-change version. the main modifications are:

1) Correction and precision on the biostratigraphy of the Montcornet borehole.
2) Removal of d18O as a paleoclimatic marker in the MS and in Fig. 11.
3) Diagenetic influence on isotopic values.
4) Modification and reinterpretation of the d13C as mentioned by the reviewers (origin of organic matter, obtusum/oxynotum negative excursion), add of a new comparative figure (figure 10).

**2) Reply to Anonymous referee #1**

**General comments.**

The research is original, novel and considered as important to the field, so it is a good candidate to be published in CP. The structure is appropriate and, in my opinion, the used language is correct. The manuscript presents a substantial contribution to scientific progress within the scope of Climate of the Past. The scientific approach and applied method referring the clay minerals are valid, but some of the isotopic data are not fully reliable and they should not be used for palaeoclimatic interpretations. The results are discussed in an appropriated way and the references are appropriated. The scientific results and conclusions are presented in a concise, clear and well-structured way, and the number and quality of figures is correct. There are no major points of conflict, as it is a high quality palaeoclimatic study mainly based on the study of clay minerals reflecting an alternation of humid and semi-arid periods during the Late Sinemurian, comparing the data obtained in two boreholes drilled in western UK (Mochras) and northern France (Montcornet). However, the isotopic data, mainly obtained from the Mochras borehole, are strongly suspicious to be strongly affected by burial diagenesis, as the δ18O presented values are too low to reflect normal seawater values and cannot be used for palaeoclimatic studies.

We thank the reviewer for agreeing to review our manuscript and for providing relevant and detailed comments. Yes, we agree with the reviewer that the oxygen isotopes are over interpreted, so we deleted this section.

Action: δ18O have been removed from MS as paleoclimatic indicator.

**Specific comments.**

Line 96. This latitude is also corroborated by the palaeomagnetic data presented by Osete et al., 2011 (Tectonophysics, 502, 105-120).

We agree. The reference is relevant.

Action: The reference has been added line 99.

Line 216.Reader has to wait until line 216 to confirm that the drill holes have recovered a supposedly continuous core of the drilled sections. That should be specified before in the text, including the core diameter and percentage of recovery of the core. Could some of the gaps found in the Montcornet hole due to the loses of core in some intervals? I assume that the hole was drilled using the wireline method, but it would be convenient to specify that in the manuscript. If the drilled section is dipping, were the thicknesses corrected respect to the depths?

The continuity of the cored sections has been specified line 105 for Mochras borehole and line 120 for Montcornet borehole. The core diameter is 85 mm in both cases and the recovery is excellent close to 100%.

Action: core diameter and the continuity of the cored sections has been added on MS lines 105 and 120.

Lines 129 to 146. It is quite singular to perform ammonite biostratigraphy in cores, due to their limited diameter, especially in the case of the Montcornet hole, were as said in line 129-130, some important hiatuses occur, and the ammonites are relatively scarce. This does not support the "High resolution data " mentioned at the beginning of the Abstract.

Yes, but the use of Yang et al 1996 and the additional determination of newly found ammonites (this study) allows to draw a suitable biostratigraphic scheme. However, the term "high resolution" has been removed. The text has been modified as in reality, the ammonites are not scarce but irregularly distributed.

Action: Irregular distribution of ammonites in Montcornet borehole has been precised in the MS line 132.

Section 4.3.1. It seems clear that the isotopic data of the study are the weakest part of the paper. Values of δ18O up to -6.54‰ reflect the presence of strong diagenetic overprints. Also the δ13C carb curve is completely different respect to the δ13C org curve, confirming the presence of the strong diagenetic overprints. As a consequence, none of the isotopic data are useful in a palaeoclimatic study (see line 324) as it is supposed to be the present paper. A diagenetic study of the carbonates is essential

to be sure that your isotopic data reflect the original Jurassic seawater conditions. Why this diagenetic study has not been performed?

Yes, we agree, and we were aware that the $\delta^{18}O$ values suffer of a diagenetic influence. δ18Ocarb and δ13Ccarb have been removed as paleoclimatic proxies.

The diagenetic study was not the objective of this work. δ18Ocarb and δ13Ccarb were not used as realistic values but as a trend, sometimes observed in other sites. The negative excursion of the *obtusum/oxynotum* zones transition is also observed in Copper Hill and Sancerre boreholes. The increase in δ18O is also an overall trend at the end of the Sinemurian. In our opinion the d18O curve is entirely shifted to low values, but the original trends are probably party preserved. However, we delete all paleoclimatic interpretations dealing with δ18Ocarb and δ13Ccarb in the new version.

Action: δ18O have been removed from MS as paleoclimatic indicator.

Line 260. "Abundant smectite indicate a limited diagenetic influence". In the paragraph above it seems that the diagenesis in the carbonates of the Mochras borehole is not negligible, but is it in the clays? Some additional justifications together with the shallow depth of burial would be welcome.

The occurrence of smectites indicates a weak clay diagenesis linked to burial. However the carbonate diagenesis is significant (nodulisation and siderite occurrence).

Action: We modify the §.

Section 5.1.2. Even of δ18O isotopic data from the Montcornet borehole are not included, it seems that diagenetic overprints are also present, showing numerous indications in the lithology. However the δ13C org curves show similar trends in both localities, indicating that this data could be reliable. The main concern is to be sure that the climatic fluctuations are not the result or are influenced by the diagenetic processes.

Yes we assume that the δ13C org curves in the two boreholes can be confidently used as climatic indicator. Contrary to δ13C carb, δ13C org seems to be less/not influenced by the diagenesis assimilar trends are observed at many other sites. We added a new figure (fig. 10) showing the correlations based on $\delta^{13}C_{org}$.

Action: We modify the MS and we add a new figure (Fig. 10). We now estimate that the negative excursion is located on *obtusum* and *oxynotum* zones.

Line 399. The ammonites zone or Zone should be uniform. Better obtusum Zone. Please check the rest of the text.

Done

Action: Done

Line 407-408. Obtusum and the oxynotum zones.

Done

Action: Done

Line 412. "Low δ18O values consistent with warm conditions". In previous sections it has been established that δ18O values cannot be used as a palaeoclimatic criteria, so it should not be used here as indicative of warm conditions, and this is contradictory with the stated in the following lines of the manuscript.

Ok, we agree.

Action: We deleted this assumption.

Lines 416-417. Reference(s) supporting the interpretation of Classopollis as an indicator of warm climate is needed. Clasopollis is a long-term pollen showing a distribution probably from the Triassic up to the beginning of the Paleogene (Vakhrameyev, 1980) surviving lots of climatic changes. So taking it as a good indicator of warming could be at least very risky, if it is not supported by more reliable data.

We agree. Correlation between *Classopollis* abundance and warm climate is risky however trends may be more suggestive. The interval corresponding to a significant increase of Classopollis in the Cleveland basin shows a potential warmer climate, supported by clay mineralogy.

Line 417-418. "Surprisingly this negative δ13C org excursion is less clearly recorded in inorganic carbon at Mochras than in the Copper Hill drillhole". This could be another indication of the strong diagenetic overprint at Mochras. That could be an indication that the multiple papers based in the isotopic signal of the Mochras borehole are values affected by the diagenetic overprint, reflecting local conditions and no global ones.

The differences can be explained by the origin of the organic matter. Higher proportion of marine organic matter during SPBE may have accentuated the negative excursion at Mochras.

Action: It has been clarified in the MS (lines 440 to 447).

Line 425-438. It would be a nice explanation but, has been compared the age of the SPBE δ13C sift with the absolute ages of the CAMP emissions? This data should be incorporated into the manuscript and supported with more data.

We agree, that a source of light carbon enrichment in relation with CAMP is theoretical. However this relationship has been discussed in detail by Ruhl et al. (2016) using an extensive bibliography on the more recent U-Pb and Ar-Ar dating available. This has been clarified in the ms.

Action: theoretical enrichment in light carbon in relation with CAMP has been clarified in the MS (lines 447 to 449).

Line 461-462. "Hot and humid interval ….. expressed by low values of δ18O". Again, this δ18O values cannot be used for palaeoclimatic interpretations.

Ok, done.

Action: δ18O have been removed from the conclusion.

**3) Reply to Anonymous referee #2**

**General comments**

This paper is an interesting contribution to the paleoclimatic reconstruction of the Early Jurassic using a multi proxy approach including Clay minerals and stable isotopes. This on 2 cores located Mochras in the Cardigan bay basin (Mochras borehole) and the Paris Basin Montcornet borehole). The topic fits therefore well with the scope of the CP journal. The paper is well written and well structured. The figures are informative and of good quality. This Ms can only be accepted only after medium to major thorough revisions, since I have some important concerns about the quality of the data and some interpretations, which are not always supported by the data.

We thank the referee for assessing our work and for providing an important review.

**Sample resolution**

In the first line of the abstract, the authors claims that it is a high resolution study (223 clay analyses). High-resolution is may be a slight overstatement, since (if we look at figures 5-6) only some 70-80 clay samples have been analysed along a 200m section at Mochras (1 sample/2.5m). The sample resolution is a little bit better in the Montcornet Borehole (around 60 samples for a 60m thick section.

The term "high resolution" has been removed.

Action: Done

**Biostratigraphy**

It looks that all the biostratigraphy is based on ammonites, it is maybe OK for the Mochras core, but not so evident for the Montcornet borehole, where several marquers are missing. It would be good to complete the biostratigraphy using nannofossils.

Nannofossils biostratigraphy is not available in Montcornet borehole but the magnetostratigraphy has been added (Yang et al., 1996 and Moreau et al 2002, this latter ref. has been added) for Montcornet (Fig.4), as it is currently used as a reference for the Sinemurian in the GTS 2020.

Action: We add magnetostratigraphy on Fig.4. We clarify the biostratigraphy on Figs. 3, 6 and 8.

At lines 128, the authors claim that the section is complicated by some important hiatuses and scarcity of ammonites. It would be important to discuss and especially locate these hiatuses.

We have modified the text dealing with the biostratigraphy based on ammonites of the Montcornet borehole.

Action: We modify the MS.

The upper Sinemurian are made of Gryphaea accumulations, probably resulting from storms interrupted by P- rich condensed levels. This makes the correlation quite difficult and some of the ammonites may be reworked.

Gryphaea accumulations are common in the Lower Sinemurian succession and rarer in Upper Sinemurian. There is however no evidence of reworked ammonites. Effectively, scattered phosphate nodules have been observed, but no P-rich condensed levels are associated with Gryphaea accumulations that could be interpreted as condensed horizons.

**Stable isotopes**

This is the weakest part of this paper. δ18O values are significantly too negative and reflects a strong diagenetic overprint. I agree that these sediments have not been too much buried, since smectite and kaolinite are still present. But it does not mean that other diagenetic processes were not acting. The presence of siderite is a good indication of a strong diagenetic process. It would have been good to analyse the bulk mineralogy by XRD (easy and fast to perform). Moreover, the most negative values of both δ18O and δ13Ccarb occur in levels, in which calcite contents are quite low (<15%). Some simple cathodoluminescence analyses would help to retrace the diagenetic story of these sediments.

Yes, we agree! $\delta^{18}O$ and $\delta^{13}C_{carb}$ have been completely removed as paleoclimatic proxies on now fig.11 and text. So we modified the text consequently. In our opinion, the $\delta^{18}O$ values are shifted to low values and $\delta^{13}C_{carb}$ values cannot be used as environmental proxies because of carbonate diagenesis. Bulk mineralogy shows indeed the occurrence of siderite (nodules observed in the core) indicating that significant carbonate diagenesis disturbed the original signal.

Action: δ18O have been removed from the MS and Fig. 11.

δ18O and δ13Ccarb can't be use for paleoclimatic reconstructions as the authors did in their figure 10 or at line 30 of the abstract. This is clearly confirmed by the observed discrepancies between the δ13Ccarb and δ13Corg.

δ18O and δ13Ccarb have been removed from now Figure 11 and their reference as a paleoclimatic marker has been removed from the abstract and conclusion.

Action: We modify the Fig. 11.

At Mochras the δ13Ccarb curve is really very different from the δ13Corg. This must be discussed in details. The δ13Ccarb shows a huge excursion in the oxynotum zone, which is not present in the δ13Corg curve.

Difference between $\delta^{13}C_{carb}$ and $\delta^{13}C_{org}$ curves seems to be related to the impact of diagenesis on carbonates. The $\delta^{13}C_{carb}$ negative excursion of the oxynotum zone is likely the result of early diagenetic

processes in this depleted carbonate interval (e.g. Ader & Javoy 1998). The $\delta^{13}C_{org}$ is probably more reliable as an environmental proxy since a similar evolution is recorded in several sedimentary basins as shown by a newly added figure (Fig. 10) of correlation that show a consistent $\delta^{13}C_{org}$ signal between UK and French basins.

Action: We modify the MS and we add a new figure (Fig. 10).

The correlation between Mochras and Montcornet based on δ13Corg curves is not convincing, since very are too many hiatuses. The authors must also explain why the δ13Corg values are more negative in the raricostatum zone of the Mochras core (down to -28) compared with coeval Montcornet values (-26). This maybe due to a difference in organic matter origin (see Schoellhorn et al, 2020 or Suan et al, 2015).

Yes we totally agree on the role of hiatuses in the Montcornet borehole (oxynatum Zone and the upper part of the raricostatum Zone – aplanatum subzone). The new figure of correlation (fig. 10) highlights the role of these hiatuses. Taking into account these hiatuses we can see that the isotopic are similar between UK and France. New data from Storm et al. (2020) indicates a potential shift in organic matter origin that may exacerbate SPBE. This point is discussed now in the MS.

Action: We add Fig. 10 and we discuss organic matter origin in the MS.

In addition, the authors may try to correlate their 13Corg curve with the one published by Peti et al, 2016, which appears to show a different trend. I suggest also to examine the δ13Corg published by Schoellhorn et al, 2020 (Dorset section), which shows several shifts in the upper Sinemurian, which can't be found neither at Mochras nor at Montcornet. Note also that Schoellhorn et al (2020,) found a negative shift in both δ13Ccarb and δ13Corg curves in the obtusum zone, confirming that the isotopic data from both Mochras and Montcornet cores are quite suspicious and can't really used for correlation. It would be good to try to correlate these isotopic records together.

It was done with the new figure 10.

**Clay minerals**

This is the most interesting part of this MS. The alternation of humid and semi-arid periods during the Late Sinemurian at Mochras is very convincing and their paleoclimatic interpretation is correct. However, it is not the case at Montcornet, where these cycles are not present. Contrary to Mochras, the kaolinite is not showing significant variations (20-30%). Since there is almost no smectite at Montcornet, I understand that the authors can't provide a SM/K ratio for that core, but they could have shown the K/I ratio, which exhibits at Mochras nice cycles showing that illite and potentially chlorite are not coupled with kaolinite, which may have originated from coeval paleosoils weathering. A different trend seems to characterize the clays distribution at Montcornet, where kaolinite, illite and chlorite shows the same trend (a simple statistic multivariate approach would be very helpful).

Yes we agree.

I am therefore not convinced that the two cores can be correlated based on clay minerals.

Yes we agree, the two boreholes cannot be correlated using clay minerals as sources are likely different.

At line 405, the authors underline the good correlation with the most prominent kaolinite increase with increased Sr ratio in the obtusum-oxynotum zones. Interestingly, this interval corresponds to very high CIA values (Schöllhorn et al,2020).

Line 408, the relationship between CIA highlighted by Schöllhorn et al. (2020) and the increase in kaolinite was added.

Action: We mention the CIA of Schöllhorn et al. (2020) in the MS.

The absence of smectite is difficult to understand and must be better explained. At line 465, the authors wrote that the different clay minerals trends may be due to the fact that Montcornet was located in a more distal location than Mochras. If it is the case, I would expect more smectite and it is really not the case. The authors linked the high amounts of smectite with sea-level low and the erosion of London-Brabant Massif. This is rather unlikely, since high smectite contents are generaly linked with high sea-level (e.g. Godet et al, 2008, Ruffel et al, 2002, Gibbs et al, 1977). Moreover, sea-level lows are characterized by a mix of clay minerals such as illite, chlorite, kaolinite..etc (Deckoninck, 1985).

We do not agree with this comment. It is true that usually the proportions of smectites are more important during periods of high sea level (e.g. Deconinck and Chamley, 1995), partly due to the differential sedimentation of clays, but on the border of the London-Brabant massif, the situation is particular. In reality, in the Jurassic (but also in the Cretaceous), this very flattened massif was very often submerged (contrary to what is indicated on most paleogeographic maps) and consequently, the clay sedimentation on its borders was the result of more distant contributions. However, during periods of low sea level, this massif had emerged and smectite pedogenesis could develop. It is clear that this massif constitutes the source of smectite. This very particular situation was highlighted in the Kimmeridgian and the Tithonian of the North-West of the Paris Basin (Boulonnais) where the lower offshore facies are rich in illite and kaolinite and devoid of smectite, while the shoreface facies are rich in smectite (see e.g., Hesselbo et al 2009). This situation is identical in the Callovian/Oxfordian on the Ardennes border (Pellenard & Deconinck, 2006) as well as in the Pliensbachien (Bougeault et al., 2017), a publication in which we explain this singularity in detail.

I suggest that the authors try to correlate their clay minerals data with the ones published by Schöllhorn et al (2020) in the Dorset. The upper Sinemurian (even if more condensed) is characterized by similar K/I and Sm/K cycles confirming that these cycles can be globally correlated and represent true paleoclimatic (semi-arid-humid) changes.

Yes, we agree, but the very different resolution of Iris Schöllhorn's study makes the comparison quite difficult. However, we added a sentence in the text indicating that the results presented in Schöllhorn et al 2020 are quite comparable with ours.

Action: Add in the MS.

**4) References added to the revised manuscript**

Franceschi, M., Corso, J.D., Cobianchi, M., Roghi, G., Penasa, L., Picotti, V., & Preto, N.: Tethyan carbonate platform transformations during the Early Jurassic (Sinemurian–Pliensbachian, Southern Alps): Comparison with the Late Triassic Carnian Pluvial Episode. Bulletin, 131(7-8), 1255-1275, https://doi.org/10.1130/B31765.1, 2019.

Moreau, M.G., Bucher, H., Bodergat, A. M., Guex, J.: Pliensbachian magnetostratigraphy: new data from Paris Basin (France). Earth and Planetary Science Letters, 203 (2), 755-767, https://doi.org/10.1016/S0012-821X(02)00898-1, 2002

Osete, M.L., Gómez, J.J., Pavón-Carrasco, F.J., Villalaín, J.J., Palencia-Ortas, A., Ruiz-Martínez, V.C., Heller, F.: The evolution of Iberia during the Jurassic from palaeomagnetic data. Tectonophysics, 502(1-2), 105-120, https://doi.org/10.1016/j.tecto.2010.05.025, 2011

Pellenard P., Deconinck J.F.: Mineralogical variability of Callovo-Oxfordian clays from the Paris Basin and the Subalpine Basin. Comptes rendus Geoscience, 338, 854-866, https://doi.org/10.1016/j.crte.2006.05.008, 2006